# *Sleeping Beauty* transposon mutagenesis identified genes and pathways involved in inflammation-associated colon tumor development

Kana Shimomura[1], Naoko Hattori[2,3], Naoko Iida[4], Yukari Muranaka[1], Kotomi Sato[1], Yuichi Shiraishi[4], Yasuhito Arai[5], Natsuko Hama[5], Tatsuhiro Shibata[5], Daichi Narushima[6], Mamoru Kato[6], Hiroyuki Takamaru[7], Koji Okamoto[8] & Haruna Takeda[1] ✉

Chronic inflammation promotes development and progression of colorectal cancer (CRC). To comprehensively understand the molecular mechanisms underlying the development and progression of inflamed CRC, we perform in vivo screening and identify 142 genes that are frequently mutated in inflammation-associated colon tumors. These genes include senescence and TGFβ-activin signaling genes. We find that TNFα can induce stemness and activate senescence signaling by enhancing cell plasticity in colonic epithelial cells, which could act as a selective pressure to mutate senescence-related genes in inflammation-associated colonic tumors. Furthermore, we show the efficacy of the Cdk4/6 inhibitor in vivo for inflammation-associated colonic tumors. Finally, we functionally validate that *Arhgap5* and *Mecom* are tumor suppressor genes, providing possible therapeutic targets for CRC. Thus, we demonstrate the importance of the inactivation of senescence pathways in CRC development and progression in an inflammatory microenvironment, which can help progress toward precision medicine.

Colorectal cancer (CRC) is the second leading cause of cancer-related death worldwide. CRC development is often initiated by acquiring loss-of-function mutations in the *APC* gene, which is observed in approximately 80% of CRC cases, followed by the accumulation of mutations in major driver genes, such as *KRAS* (~40%), *SMAD4* (~10%), and *TP53* (~50%), leading to malignant progression[1]. Chronic inflammation is known to promote CRC development and progression, and a subtype of CRC characterized by high inflammatory signatures correlates with poor prognosis[2,3]. Patients with inflammatory bowel disease (IBD) are

at a higher risk of developing CRC[4–6]. Genomic analysis of colitis-associated cancers (CAC) has shown that *TP53* is the most commonly mutated gene in CAC[7–10]. In addition, several genes, such as *SOX9*[7], *EP300*[7], *ARID1A*[9], *CDH2*[9], and *FBXW7*[10] were more frequently mutated in CAC. These data suggested that the genetic mutational profile of CAC differs from that of sporadic CRC[7–10].

To comprehensively understand the genes and pathways involved in inflammation-associated CRC, we performed *Sleeping Beauty* (SB) transposon screening in a mouse model of colitis. SB transposon

[1]The Laboratory of Molecular Genetics, National Cancer Center Research Institute, Tokyo, Japan. [2]Division of Epigenomics, National Cancer Center Research Institute, Tokyo, Japan. [3]Department of Epigenomics, Institute for Advanced Life Sciences, Hoshi University, Tokyo, Japan. [4]Division of Genome Analysis Platform Development, National Cancer Center Research Institute, Tokyo, Japan. [5]Division of Cancer Genomics, National Cancer Center Research Institute, Tokyo, Japan. [6]Division of Bioinformatics, National Cancer Center Research Institute, Tokyo, Japan. [7]Endoscopy Division, National Cancer Center Hospital, Tokyo, Japan. [8]Advanced Comprehensive Research Organization, Teikyo University, Tokyo, Japan. ✉e-mail: hartaked@ncc.go.jp

screening is a powerful genetic tool for genome-wide identification of cancer driver genes and for understanding the evolutionary forces that promote cancer progression[11–14]. SB transposons continuously introduce insertional mutations so that cells with genetic mutations that are best adapted to the cancer microenvironment continue to be selected. Previously, we performed SB screening using different sensitizing mutations and identified 1333 candidate cancer driver genes (CCDGs) involved in intestinal tumor development. By comparing CCDGs to human datasets, we enriched potent cancer driver genes based on cross-species comparisons and functionally validated *Zfp292* as a tumor suppressor. An important insight gained from our previous study was how much the order in which the cancer-causing mutations occur matters[15,16], providing information regarding evolutionary pressures driving CRC development.

In the present study, we identify 142 genes that are frequently mutated in inflammation-associated tumors, including senescence and activin signaling genes. Interestingly, we find that the inflammatory cytokine TNFα simultaneously induces stemness and senescence signaling activation in colonic epithelial cells, and such seemingly paradoxical pathway activation can be a selective pressure for cancer development. Thus, we have provided datasets that will help develop different personalized therapies for treating inflammation-associated CRC and a concept that cellular plasticity in the inflammatory microenvironment can act as an evolutionary pressure driving CRC development.

## Results

### Colitis reduced the survival of mice and promoted large tumor development

To identify the genes involved in inflammation-associated colonic tumor development, we performed SB mutagenesis screening in a colitis model using SB11 knock-in mice[13], T2/Onc2 transposon transgenic mice[17], and Villin-CreERT2 transgenic mice[18], which express intestinal epithelial-specific Cre. To promote tumor development, we introduced CRC driver mutations; *KrasG12D*[19] or *Tgfbr2flox*[20] or *Trp53R270H*[21] or a combination of the two mutations (Fig. 1a); *TP53* and *KRAS* mutations were observed in colitis-associated tumors[8], and loss of the TGFβ type II receptor was shown to be involved in colitis-associated tumor development[22]. We activated SB mutagenesis specifically in intestinal epithelial cells by tamoxifen administration at approximately 4 weeks of age, and then treated the mice with three rounds of 2.5% Dextran Sodium Sulfate (DSS) from 10 weeks of age to induce colitis (Fig. 1b). DSS-treated mice are shown as +DSS (e.g., K-SB mice treated with DSS are shown as K-SB + DSS). We did not include DSS-treated SB negative controls because the number of tumors in DSS-treated mice without carcinogens was relatively low[23,24].

Mouse survival was significantly shortened by DSS compared to non-DSS controls in the K-SB + DSS, KT-SB + DSS, and KP-SB + DSS groups (Fig. 1c and Supplementary Fig. 1a). Large colonic tumors (≥5 mm) were frequently observed in K-SB + DSS and KT-SB + DSS, mice, whereas no large tumors were developed in their controls, suggesting that colitis promoted large tumor development. In KP-SB + DSS and SB + DSS mice, the total number of colonic tumors significantly increased compared to that in the controls; however, P-SB + DSS mice did not exhibit significant differences (Fig. 1d and Supplementary Fig. 1b). Histopathological analyses of the tumors (≥3 mm) showed that invasive adenocarcinomas developed frequently (Fig. 1e–i and Supplementary Fig. 1c–k). The percentages of adenocarcinomas were 24.1%, 33.3%, 56%, and 33.3% in the tumors of K-SB + DSS, KT-SB + DSS, KP-SB + DSS, and P-SB + DSS mice, respectively (Fig. 1j). We also found two cases of metastasis in KT-SB + DSS mice (Fig. 1j and Supplementary Fig. 1l–q), suggesting that *Tgfbr2* loss may promote metastasis, although the frequency was low. These data show that colitis shortened the mouse survival, and promoted the development of large tumors in the presence of SB insertional mutations.

### Identification of candidate cancer driver genes

We established an in-house informatics pipeline to detect common transposon integration sites (CISs) and identified 1,459 candidate cancer driver genes (CCDGs) from 315 tumors in eight screens: K-SB, KT-SB, KP-SB, and P-SB with or without DSS (Supplementary Data 1). Of these, we identified 718 genes, including *Cdkn2a, Smad2*, and *Nf1* (Fig. 2a and Supplementary Data 2). Previous SB screening for CRC genes has been performed in tumors of the small intestine and colon[13,14,16]; however, this study only focused on colon tumors, which could provide more reliable datasets of candidate CRC genes.

### Frequent mutations in *Cdkn2a* and *Trp53* in inflammation-associated tumors

To identify genes that were more frequently mutated in DSS tumors, we compared the insertional mutation frequencies between DSS and non-DSS tumors (Supplementary Fig. 2) and enriched 142 genes as "inflammation-associated genes" (Supplementary Data 3). Pathway and gene ontology (GO) analyses showed that TGFβ signaling (*Acvr1b, Acvr2a*, and *Smad4*), adherens junction (*Ctnna1* and *Ctnnd1*), senescence (*Trp53* and *Cdkn2a*), chromatin remodeling (*Chd7, Arid1a*, and *Kdm5a*), and RNA splicing (*Hnrnpm* and *Rbm5*) genes were highly enriched (Fig. 2b). Cancer genome sequencing studies of human CACs have identified mutations in *TP53* and *ACVR2A*[7,8,25] as commonly mutated genes in CAC, which were also identified in our screens as inflammation-associated genes.

To gain a comprehensive understanding of the insertional mutation profiles influenced by the microenvironment or preexisting driver mutations, we generated an oncoplot (Fig. 2c). The percentage of tumors carrying insertional mutations in the Wnt signaling gene was 70–100% in each screen, which was not different between the analyzed groups. The frequency of insertional mutations in adherens junction genes was also not different between tumors that were treated with DSS and those that were not; however, *Arhgap5* was more commonly mutated in KP-SB + DSS tumors (Supplementary Fig. 2). In contrast, the number of tumors carrying insertional mutations in senescence and chromatin remodeling genes was significantly more frequent in DSS tumor than in no DSS tumor (*P* = 0.0063 and 0.0003, respectively, by two-sided Fisher's exact test). Remarkably, inflammation-associated tumors in the K-SB + DSS, KT-SB + DSS, and KP-SB + DSS mice strongly selected mutations in *Cdkn2a* and *Trp53*; however, the selected genes differed. Insertional mutations in *Cdkn2a* were preferred in K-SB + DSS and KT-SB + DSS, whereas insertional mutations in *Trp53* were preferred in KP-SB + DSS, which likely disrupted the remaining wild-type (wt) allele of *Trp53* (Fig. 2d and Supplementary Fig. 2b).

To focus on how *Cdkn2a* mutations are involved in the early stages of tumor development, insertional mutation frequencies in nascent tumors (< 2 mm) were compared. Interestingly, in nascent tumors, the frequency of *Cdkn2a* mutations was significantly higher in DSS tumors (Fig. 2e). Furthermore, the proportion of tumors with *Cdkn2a* mutations increased in adenocarcinomas (Fig. 2f). These data suggest that colonic cells carrying a KrasG12D mutation selectively acquire *Cdkn2a* mutations to promote tumor development in the inflammatory microenvironment.

### Activation of senescence signaling by TNFα

In DSS colitis, epithelial damage and repair were repeatedly induced (Fig. 3a and Supplementary Fig. 3a–c), providing a unique microenvironment for tumor development. To elucidate the molecular mechanisms underlying the frequent mutation of *Cdkn2a* in the early stage of DSS tumor development, we focused on changes in *Cdkn2a* expression, which is a marker for senescence signaling. Interestingly, the expression of p16/p19 (both encoded by *Cdkn2a*) increased in the colon tissue of DSS-treated mice (Supplementary Fig. 3c), and in colonic epithelial-derived organoids established from DSS-treated

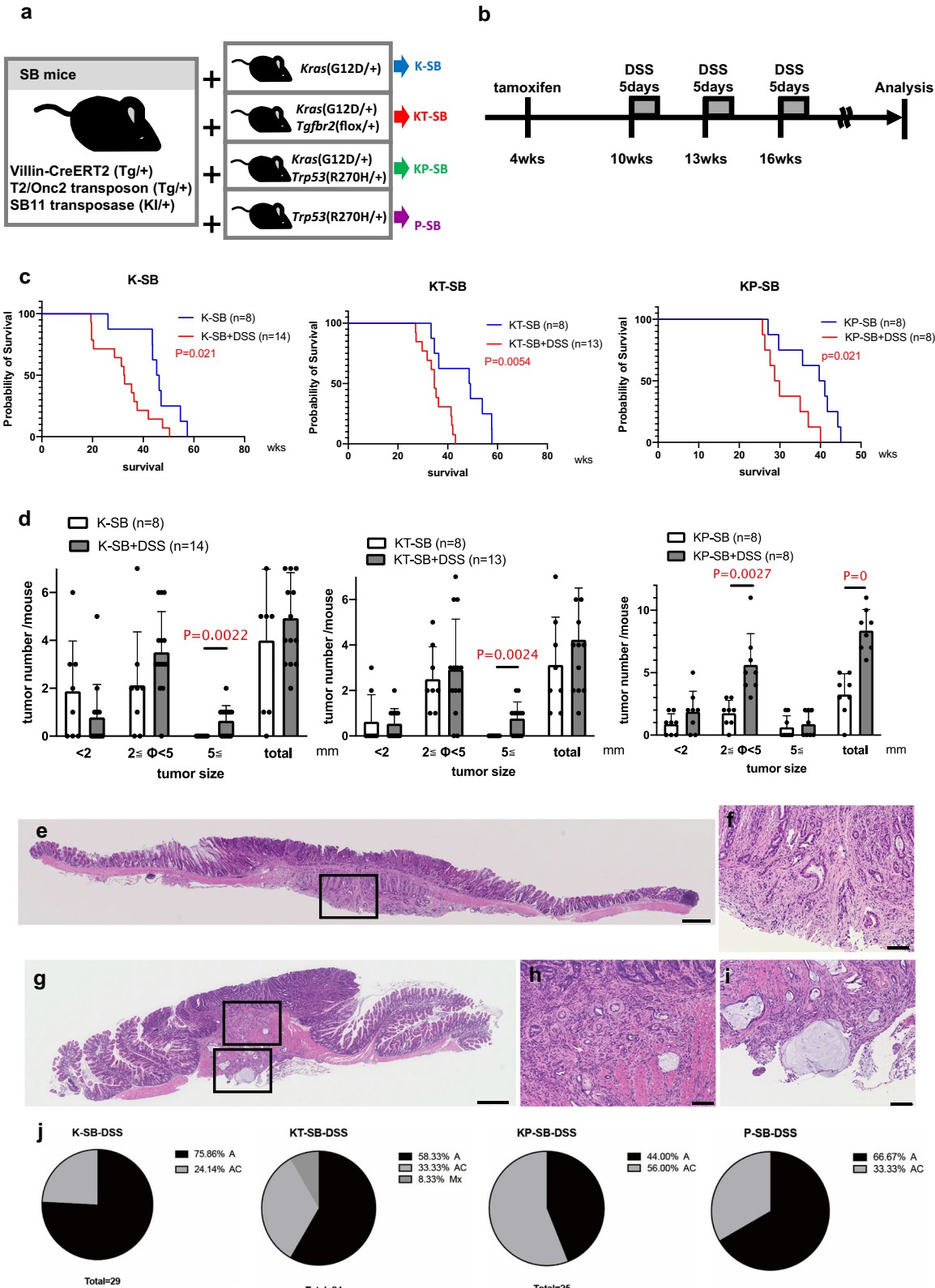

**Fig. 1 | DSS accelerated tumorigenesis and shortened lifespan. a** The alleles used for each SB mutagenesis screening and abbreviations for each mouse group. **b** Tamoxifen (20 mg/ml) was administered intraperitoneally (i.p.) three times to activate Cre at 4 weeks of age, and then 2.5% DSS in drinking water was administered three times for 5 days to induce chronic inflammation in the colon. **c** Survival of mice was significantly shortened by DSS treatment. The average age of K-SB (45.5 weeks), K-SB + DSS (32.4), KT-SB (46.4), KT-SB + DSS (35.2), KP-SB (38.0) and KP-SB + DSS (31.3) mice. Log-rank test. **d** The number of colonic tumors/mouse for each screen. Data are represented as mean values +/− SD. Two-sided *t* test. **e–i** HE staining for representative colon tumors of K-SB + DSS mice (**e**, **f**), KT-SB + DSS mice (**g–i**). A High-magnification image of the inset in (**e**) is (**f**). High-magnification images of the insets in (**g**) are (**h**) and (**i**). Scale bars, 500 μm (**e**, **g**) and 100 μm (**f**, **h**, **i**). **j** Pie charts showing the percentage of adenomas (A), adenocarcinomas (AC), and metastatic tumors (Mx).

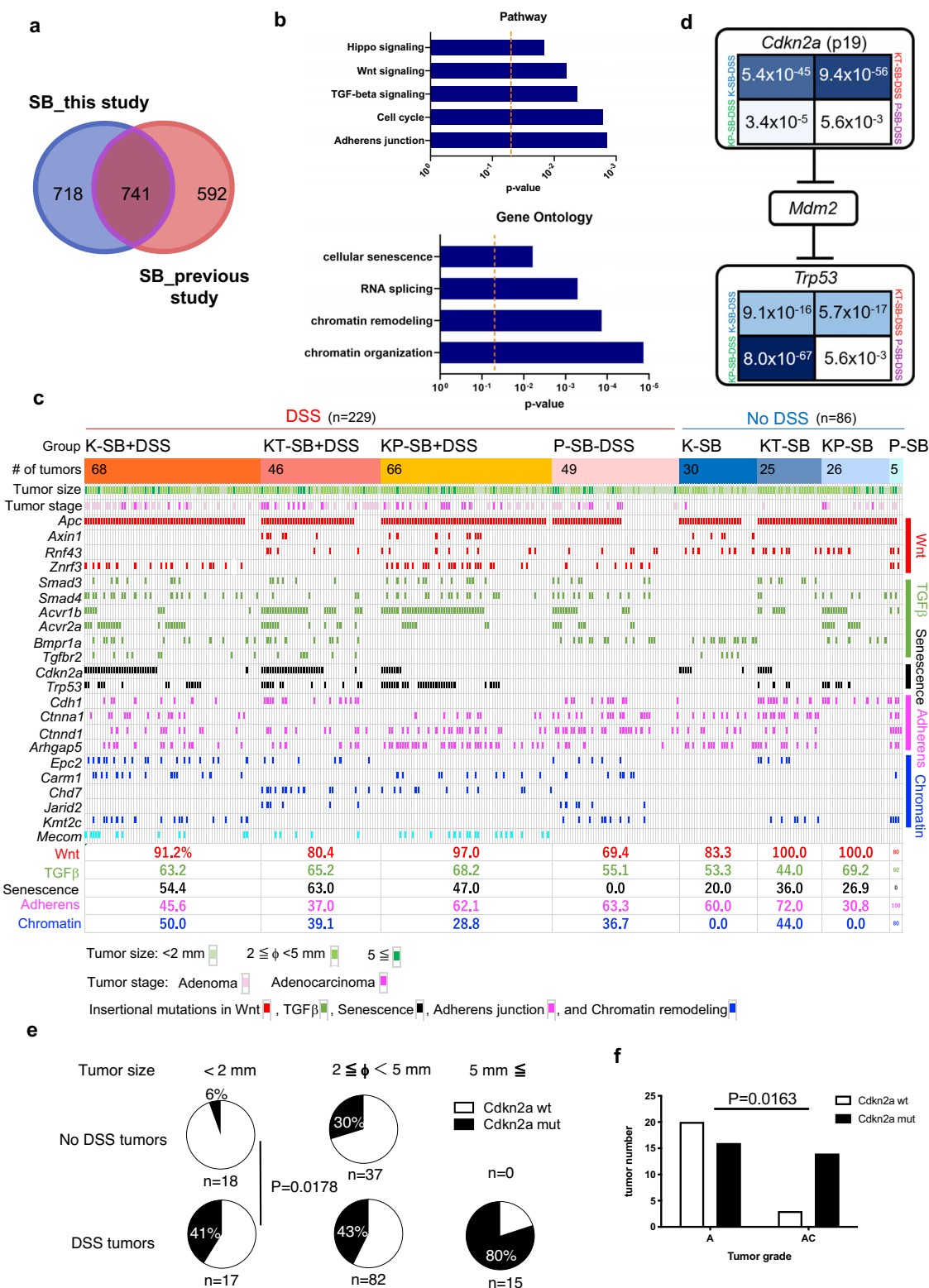

**Fig. 2 | Comprehensive analyses of CCDGs. a** Venn diagram showing the number of overlapping CCDGs for CRC identified from this study and previous study[16]. **b** Enriched pathways for genes frequently mutated in DSS tumors. Two-sided Fisher's exact test. **c** The oncoplot shows the frequency of insertional mutations in genes involved in indicated signaling pathways and functions. **d** Comparison of frequencies of tumors carrying CISs in *Cdkn2a* and *Trp53* among 3 screens, two-sided Fisher's exact test. **e** Pie charts showing the percentages of tumors carrying *Cdkn2a* insertional mutations in DSS tumors and no DSS tumors by size, two-sided Fisher's exact test. **f** Bar graphs showing the number of tumors carrying *Cdkn2a* insertional mutations in adenomas (A) and adenocarcinomas (AC), two-sided Fisher's exact test.

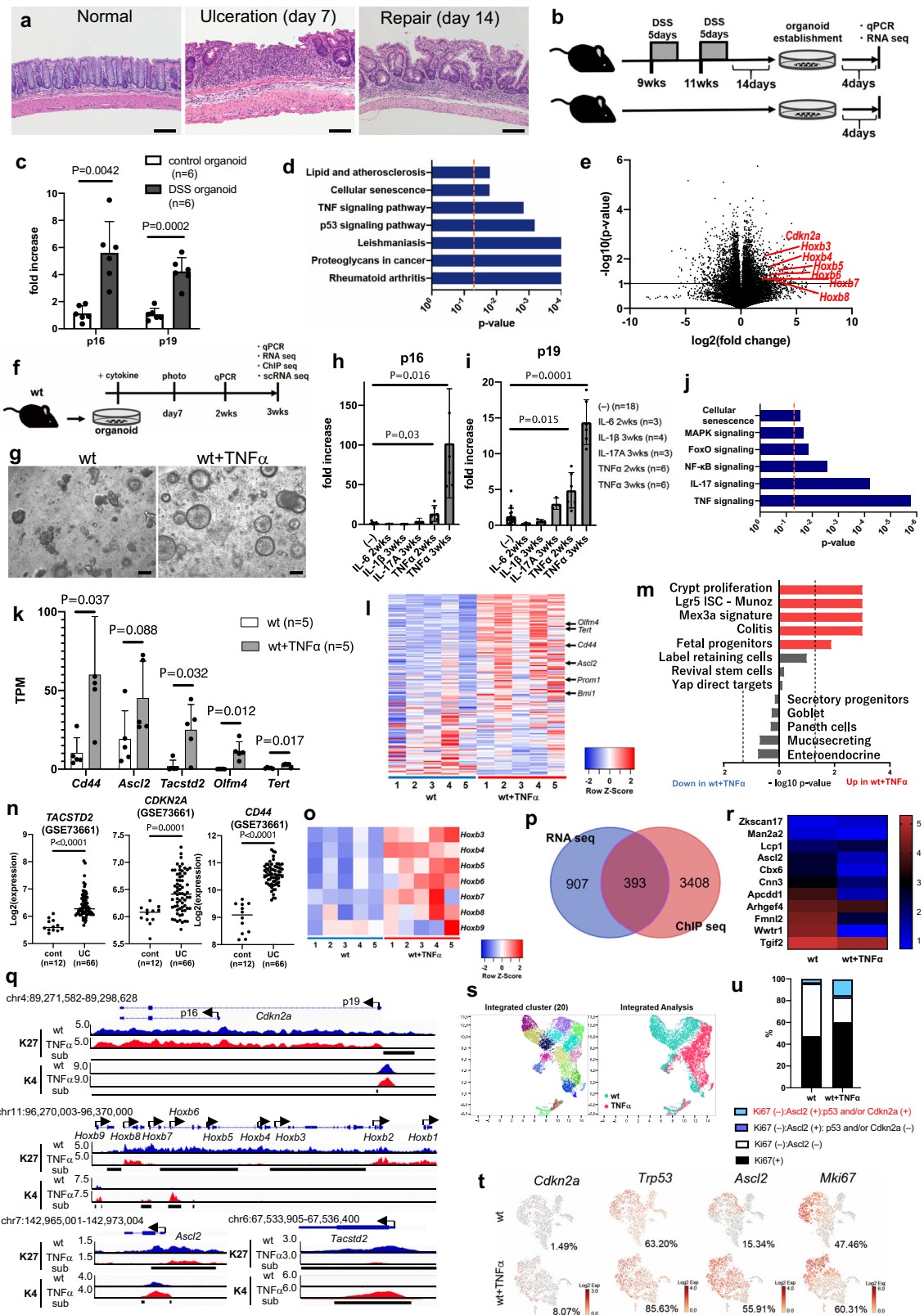

mice (DSS organoids, Fig. 3b, c). Next, we performed RNA-seq using DSS and control organoids and found different expression profiles as characterized by the activation of senescence signaling in DSS organoids (Fig. 3d and Supplementary Fig. 3d, e). Notably, several *Hoxb* genes were upregulated in the DSS organoids (Fig. 3e and Supplementary Fig. 3f). The increased expression of *Hoxb* correlated with the demethylation of histone H3 lysine-27 trimethylation (H3K27me3) in

the promoter[26], implicating a change in histone modifications in DSS organoids.

To identify the factors inducing senescence signaling in colitis, we treated wt colonic organoids with various inflammatory cytokines that were upregulated in the colons of DSS-treated mice (Supplementary Fig. 3c) and patients with UC[27]. Interestingly, p16/p19 expression increased when TNFα was added (Fig. 3f–i). Using transcriptome

**Fig. 3 | TNFα-induced p16/p19 expression and colonic stem cell markers were associated with epigenetic remodeling. a** HE for the normal mouse colon, the ulcer at day 7 from DSS, and regenerating epithelia. Bars; 100 μm. **b** An experimental design to establish organoids from DSS-treated mice. **c** Quantitative PCR (qPCR) results for p16 and p19 using wt (control) and DSS organoids. Six independent samples. **d** A pathway analysis showing deregulated pathways in DSS organoids. Two-sided Fisher's exact test. **e** A volcano plot showing the expression changes in DSS compared to control organoids. Three mice were used for control and DSS, respectively. **f** A picture showing the timing of cytokine addition and sample collection. **g** Photos of wt and wt+TNFα organoids. Bars; 200 μm. **h, i** qPCR showing expression changes by cytokines in wt organoids for p16 (**h**), p19 (**i**). **j** A pathway analysis for RNA-seq using wt and wt+TNFα organoids. Two-sided Fisher's exact test. **k** RNA-seq for colonic stem cell genes[29] using wt and wt+TNFα organoids. **l** A heatmap for the colonic stem cell genes[29] obtained by RNA-seq between wt and wt+TNFα organoids. **m** GSEA analysis for RNA-seq. Two-sided Fisher's exact test. **n** Graphs showing RNA expression of indicated genes in colonic mucosa of the healthy control (cont) and UC patients (UC) derived from the dataset. **o** A heatmap showing the expression of genes in the *Hoxb* cluster obtained by RNA-seq between wt and wt+TNFα organoids. **p** Venn diagram showing the number of genes identified from ChIP-seq and RNA-seq. **q** IGV images for ChIP-seq in indicated genetic loci. Unique peaks subtracted (sub) were shown by bold black lines. K27:H3K27me3, K4:H3K4me3. **r** A heatmap showing the intensity of H3K27me3 in promoters of stem cell genes with increased expression in wt+TNFα organoids. **s** Integrated analysis for cell clustering by scRNA-seq. **t** T-SNE clustering analyses showing cell populations. **u** Bar graphs showing % of cells double positive for a stem cell marker and senescence marker(s) in Ki67(−) cells. Data are presented as mean values +/− SD, and tested with two-sided *t* test for (**c, e, h, i, k, n**).

analysis, we also observed the upregulation of senescence signaling genes such as *Cdkn2a, Mapk11, Mapk12,* and *Slc25A4* in TNFα-treated organoids by, clearly showing that TNFα could induce senescence signaling in the colonic epithelial cells (Fig. 3j).

## Simultaneous induction of stem cell markers and senescence genes by TNFα via epigenetic remodeling

The organoids became round in the presence of TNFα (Fig. 3g and Supplementary Fig. 3g), which was associated with increased stemness[28]. Expression of colonic stem cell markers, *Cd44, Ascl2, Olfm4,* and *Tert,* was induced by TNFα (Fig. 3k), and the colonic stem cell genes[29] enriched in TNFα-treated organoids (Fig. 3l). *Tacstd2,* a marker of regenerating colonic epithelial cells[30], was also upregulated. The increased expression of colonic stem cell markers is referred to as "stemness" in this study, indicating that TNFα can induce stemness in colonic organoids. To further determine which cell types in the intestine are more similar to TNFα-treated organoids, we compared our RNA-seq dataset with previously reported gene sets[30,31]. Interestingly, the cell state in the TNFα-treated wt organoids was similar to *Mex3a* positive cells, *Lgr5* positive cells, and colitis cells, but not correlated with label-retaining cells (LRC) or other differentiated cells. These data also supported our finding that TNFα could induce stemness in the colonic epithelial cells (Fig. 3m). In the colonic mucosa of patients with UC, characterized by excess TNFα expression, the expression of *CDKN2A, CD44,* and *TACSTD2* was significantly increased compared to that in healthy controls (GSE87466 and GSE73661), which is consistent with our findings (Fig. 3n and Supplementary Fig. 3H).

Similar to DSS organoids, the expression of the *Hoxb* cluster genes also increased in TNFα-treated organoids (Fig. 3o), implicating the epigenetic change by TNFα. To confirm this, we performed ChIP-seq for H3K27me3, a marker of transcriptional repression[32], and for H3K4me3, a marker of transcriptional activation[33]. The total number of genes predicted to increase the transcription in TNFα-treated organoids was 3801 (Fig. 3p and Supplementary Data 4), in which "senescence" and "cell cycle" genes were included (Supplementary Data 5). Note that we focused on genes upregulated by TNFα By comparing RNA-seq and ChIP-seq results, we identified 393 genes whose elevated RNA expression was consistent with epigenetic changes (Supplementary Data 6). These included senescence-associated genes (*Cdkn2a, Mapk11,* and *Mapk12*), colonic stem cell markers (*Tacstd2* and *Ascl2*), and *Hoxb* cluster genes (Fig. 3q and Supplementary Fig. 4a). Although genome-wide histone modifications did not differ with TNFα (Supplementary Fig. 4b, c), the intensity of H3K27me3 in the promoters of stem cell genes with increased expression was significantly reduced (Fig. 3r), indicating epigenetic remodeling by TNFα.

To analyze the gene expression profile at the single-cell level, we performed scRNA-Seq. The integrated analysis showed that cell clusters distinct from control organoids appeared in the TNFα-treated organoids (Fig. 3s). TNFα increased *Ascl2*⁺ or *Cd44*⁺ cell populations (*Ascl2*⁺:3.6 times, *Cd44*⁺:4.4 times), and decreased the *Cck*⁺ enteroendocrine cell population, and the *Muc2*⁺ goblet cell population (*Cck*⁺:0.3 times, *Muc2*⁺:0.7 times), showing the expansion of stem cells in TNFα-treated organoids (Fig. 3t and Supplementary Fig. 5a). However, the proportion of *Mki67*(−) cells did not decrease, as expected from the increase in the stem cell population (Fig. 3t). This is probably because TNFα contributes to the expansion of *Mki67* (-): *Ascl2*(+) non-dividing stem cells. Interestingly, the induction of senescence genes (*Trp53* or *Cdkn2a* or *Cdkn1a*) in the *Mki67* (−): *Ascl2*(+) non-dividing stem cells was observed 4.3 times more in TNFα-treated organoids compared to that in control organoids (Fig. 3u and Supplementary Fig. 5). Because these non-dividing stem cells can potentially proliferate, disruption of the senescence pathway may cause these cells to actively divide in the presence of TNFα.

## Cells acquiring *Cdkn2a* or *Trp53* mutations gained a growth advantage in the presence of TNFα

To examine whether TNFα exerts similar effects in a KrasG12D background, we established colonic organoids expressing KrasG12D (Fig. 4a, b and Supplementary Table 2). While KrasG12D expression itself did not induce the expression of colonic stem cell markers or *Hoxb* cluster genes (Supplementary Fig. 6a), TNFα induced the expression of p16/p19, stem cell markers and the *Hoxb* cluster genes (Fig. 4c–f). Transcriptome analysis showed that TNFα deregulated cell senescence signaling, p53 signaling, and cell cycle control (Fig. 4g and Supplementary Fig. 6b), but did not affect organoid proliferation (Fig. 4h). TNFα could positively regulate the cell cycle, as was illustrated by the upregulation of Cdks and Cyclins, and negatively, as indicated by the activation of the cell senescence pathway activation (Fig. 4i). Therefore, in the TNFα-rich environments, once cancer cells acquire *Cdkn2a* or *Trp53* mutations, the cells can gain growth advantage. To test this hypothesis, we introduced mutations in *Cdkn2a* or *Trp53* in KrasG12D colonic organoids, and compared cell viability (Fig. 4j, k and Supplementary Fig. 7a, b) and cell proliferation (Fig. 4l, m and Supplementary Fig. 7c) with or without TNFα. Cell viability and proliferation were significantly increased in the presence of TNFα, showing that the cell cycle was promoted by TNFα when *Cdkn2a* or *Trp53* was mutated on the KrasG12D organoids.

To confirm that TNFα induces *CDKN2A* expression in human colonic cells, we used CRC organoids (CRC20) carrying mutations in *APC* but wt for *TP53* and *KRAS*, and normal cecum epithelial organoids (Ce) and confirmed the induction of p14 by TNFα in both cell types (Fig. 4n, o). To gain more comprehensive insights, we examined whether *TNF* overexpression/amplification was associated with *TP53* mutations in the TCGA dataset using Pan-cancer analyses[34]. Cancer tissues with *TNF* overexpression/amplification co-occurred with mutations in *TP53* (*P* = 0.0442), raising the possibility that increased levels of TNFα are associated with *TP53* mutations in human cancers (Fig. 4p). Alternatively, IFNγ overexpression/

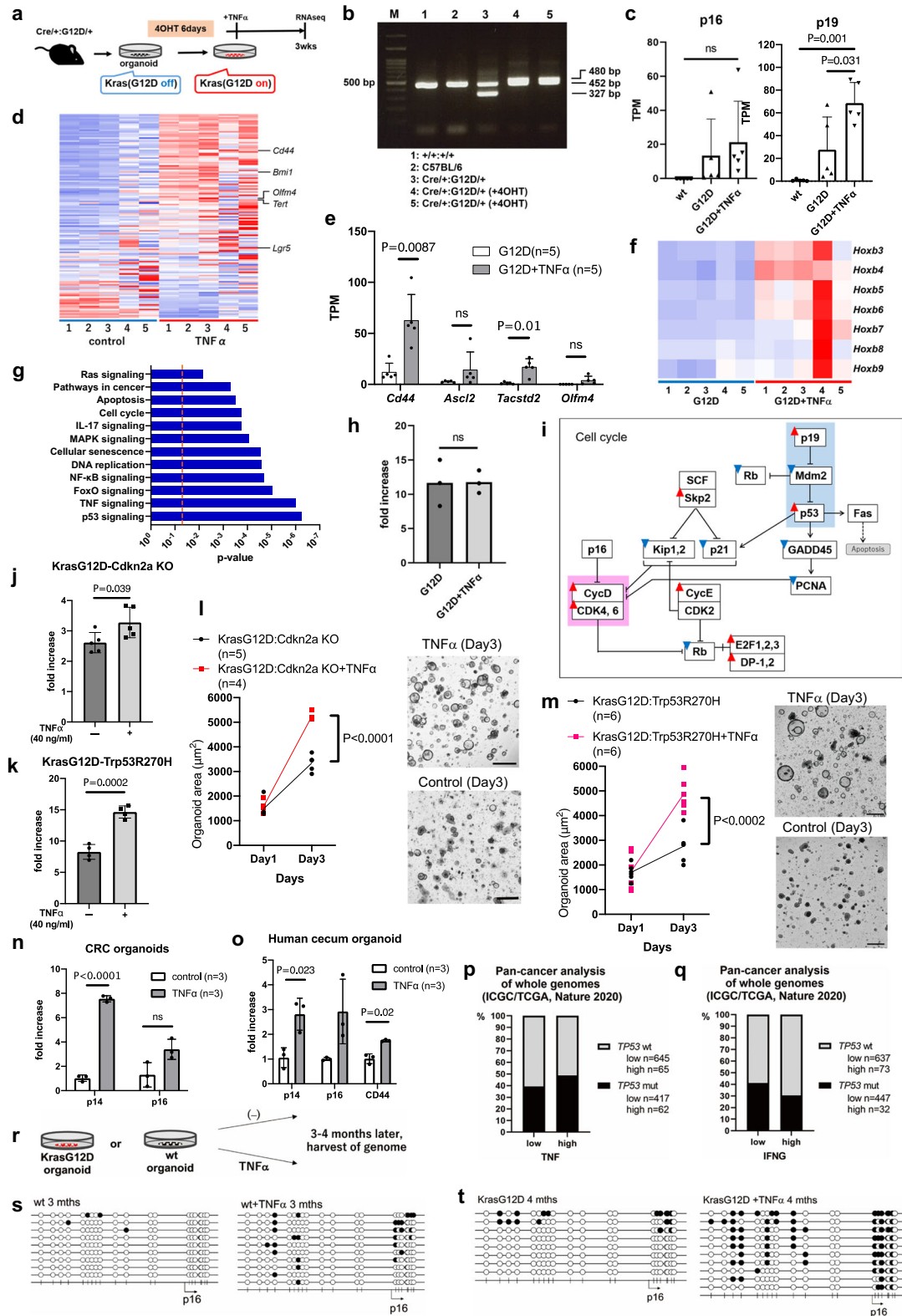

amplification did not cooperate with *TP53* mutations (a tendency for mutual exclusion, *P* = 0.0368) (Fig. 4q).

In human colitis-associated tumors, inactivation of *CDKN2A* appears to be caused by DNA methylation rather than genetic mutations[35–38]. Spontaneous DNA hypermethylation in the promoter region was found to occur during chronic inflammation[39]. Therefore, we examined whether TNFα induces DNA methylation in the *Cdkn2a* region in vitro. We exposed organoids with TNFα for ≥3 months and

analyzed the methylation status (Fig. 4r). Remarkably, the level of DNA methylation was increased by TNFα in both wt and KrasG12D organoids (Fig. 4s, t). The expression of DNA methyltransferases was slightly increased by TNFα in colonic organoids (Supplementary Fig. 8a), raising the possibility that TNFα signaling directly promotes DNA methylation by positively regulating DNA methyltransferases. Alternatively, cells with naturally occurring DNA methylation could be selected under prolonged TNFα exposure to gain proliferative ability.

**Fig. 4 | TNFα can simultaneously induce stemness and senescence markers.**
**a** An establishment of KrasG12D organoids. **b** Detection of recombind (480 bp), wt (452), and knock-in allele (327) of the Kras locus by PCR. **c** Bar graphs showing p16/p19 expression in wt, KrasG12D (G12D) and G12D + TNFα organoids. Five independent samples. **d** A heatmap for the colonic stem cell genes[29] between G12D and G12D + TNFα organoids. **e** Bar graphs showing expression changes for colonic stem cell genes using G12D and G12D + TNFα organoids. Five independent samples. **f** A heatmap showing differentially expressed genes in the *Hoxb* cluster between G12D and G12D + TNFα organoids. **g** Bar graphs for pathway analysis for differentially expressed genes between G12D and G12D + TNFα organoids. Two-sided Fisher's exact test. **h** Organoid proliferation in G12D and G12D + TNFα organoids. Mean values +/− SEM, two-sided *t* test, three independent samples. **i** Genes in cell cycle regulations. Red rectangles: increased expression in G12D + TNFα. Blue rectangles: decreased in G12D + TNFα. Genes highlighted in light blue are likely to inhibit the cell cycle, whereas those highlighted in pink are likely to promote the cell cycle.

**j** Organoid viability (Day3/Day1) for KrasG12D:Cdkn2aKO organoids with or without TNFα. Five independent samples. **k** Organoid viability (Day3/Day1) for KrasG12D:Trp53R270H organoids. Four independent samples. **l** Quantification of organoid growth for KrasG12D:Cdkn2aKO organoids. Five (control) and four (+ TNFα) independent samples. The full-focus images of organoids. Bars; 0.5 mm. **m** Quantification of organoid growth for KrasG12D:Trp53R270H organoids. Six independent samples. The full-focus images of organoids. Bars; 0.5 mm. **n, o** Bar graphs showing expression changes in human CRC-derived organoids (CRC20) or human normal cecum organoids with or without TNFα (**o**). **p, q** The percentage of *TP53* mutations in *TNF* low or high (**p**), and *IFNG* low or high (**q**) tumors in the TCGA dataset. **r** A picture describing the experimental design. **s** Bisulfite sequencing for the *Cdkn2a* locus in wt organoids and wt+TNFα cultured for 3 months. **t** Bisulfite sequencing for *Cdkn2a* locus using G12D and G12D + TNFα organoids cultured for 4 months. Data are presented as mean values +/− SD, tested with two-sided *t* test for (**c**, **e**, **j**–**o**).

Thus, we could present a possible mechanism of how *Cdkn2a* was inactivated in inflammation-related cancer development.

## A selective CDK4/6 inhibitor responded to inflammation-associated colon tumors in mice
Recently, selective CDK4/6 inhibitors (CDK4/6i) have shown remarkable efficacy in treating HR + / HER2- advanced breast cancer[40,41]. Moreover, several clinical trials of selective CDK4/6i for advanced solid cancers are ongoing[40], as genetic alterations predicted to drive the hyper-activation of CyclinD-CDK4/6 are common in various cancers.

Motivated by our results and previous reports, we tested palbociclib in vitro using mouse tumor organoids carrying loss-of-function mutations in *Apc* (*Apc*^Δ716/+^), an activating *KrasG12D*^/+^ mutation (hereafter referred to as AK organoids), and AK-Cdkn2a KO organoids. We confirmed that palbociclib affected both cell types (Fig. 5a). Palbociclib was then administered to the K-SB + DSS mice (Fig. 5b). K-SB + DSS mice treated with palbociclib lived longer than control mice (Fig. 5c), demonstrating the efficacy of palbociclib in inflammation-associated tumors in mice. To identify candidate genes involved in palbociclib resistance, we analyzed the tumor genomes from K-SB + DSS and K-SB + DSS+Palbociclib mice. Interestingly, the genes that were frequently mutated in K-SB + DSS+Palbociclib tumors included *Trp53* and *Cdh1* (Fig. 5d). However, the role of p53 in the CDK4/6i-induced growth arrest remains unclear. While CDK4/6i can induce growth arrest in malignant cells lacking functional p53[40], *TP53* mutations were strong predictors of CDK4/6i resistance in a panel of 560 cancer cell lines[42]. Our in vivo data showed that cancer cells preferentially gained mutations in *Trp53* to acquire palbociclib resistance.

## Organoids with *Cdkn2a* mutations became resistant to Activin-induced growth arrest
In DSS tumors, activin signaling genes (*Acvr1b*, *Acvr2a*, and *Smad4*) were highly mutated (Supplementary Data 2 and Fig. 2c), suggesting the involvement of activin signaling in inflammation-associated tumor development. Activin expression is increased in fibroblasts[43] of the colonic mucosa of patients with UC. Previously, we showed that *Acvr1b* and *Acvr2a* function as tumor suppressors in the colon, and their ligand, activin, induces growth arrest in colonic organoids[44]. Therefore, inactivation of activin receptors may be a strategic approach for cancer cells to become resistant to activin-induced growth arrest in the inflammatory microenvironment.

We found that the expression of *Inhba*, which encodes activin, was increased in tumor tissues compared to that in non-tumor tissues in K-SB + DSS and KT-SB + DSS mice (Fig. 5e), indicating that cancer cells were continuously under selection pressure to gain mutations in the activin receptors. We also confirmed the increased expression of *Tgfb1* and *Tnf* in tumor tissues. Interestingly, AK-Cdkn2aKO organoids were resistant to activin-induced growth arrest (Fig. 5f). These

data suggest that inactivation of *Cdkn2a* may confer cancer cells with the ability to survive in an activin-rich inflammatory environment (Fig. 5g).

## Prioritization of CCDGs via comparative genomics
Mutations commonly found in mice and humans are likely to be potent driver genes because their functions in cancer are likely conserved across species. We compared 1,459 CCDGs identified by SB screening with two human datasets to prioritize CCDGs as potent cancer driver genes. The number of overlapping genes between CCDGs and genes mutated in ≥1.5% of TCGA-CRC[34] was 812 (Supplementary Data 7), and the number of overlapping genes with OncoKB[45] genes was 200 (Supplementary Data 7). The overlaps were statistically significant ($P < 0.0001$ and $P = 0.014$, respectively), and these overlapping genes may be potent cancer driver genes. To further prioritize, we combined genes with mutation frequencies ≥3% of the TCGA-CRC with OncoKB genes and compared them to 1,459 CCDGs to enrich 471 genes, which included known CRC genes such as *APC*, *SMAD4*, *TP53*, and *RNF43*, and genes whose functions in CRC were unknown, such as *IFNGR1* and *TRIM33* (Fig. 6a and Supplementary Data 2 and 7). Similarly, we compared 142 inflammation-associated tumor genes with human genes (≥3% of TCGA-CRC + OncoKB). Overlapping genes included known CAC genes, such as *TP53*, and genes with unknown functions in CRC and CAC, such as *ARHGAP5* and *MECOM* (Fig. 6b and Supplementary Data 3), which should be analyzed with the highest priority.

## Functional validation of CCDGs
Genome information alone cannot prove whether a gene functions as a driver gene, and proof-of-concept experiments are necessary. Based on prioritization analyses using comparative genomics, we focused on *Arhgap5* and *Mecom*. Transposon insertions appeared to inactivate these two genes (Fig. 6c) and termination mutations were observed in human CRC (Fig. 6d), suggesting that these genes could function as tumor suppressors. *MECOM* encodes a transcriptional regulator protein, and 7.6% of CRC with MSI-H[46] carry frameshift mutations in *MECOM*. MECOM acts as a negative feedback regulator of NFκB[47], raising the possibility that loss of function mutations in *Mecom* can activate NFκB signaling to promote the cell cycle in the inflammatory microenvironment.

The standard approach to validate the function of cancer drivers is to create genetically engineered mice and evaluate whether these mice promote tumorigenesis in vivo; however, this approach is time-consuming and not ideal for validating several genes. Although human-derived cell lines can be used for validation, the reproducibility of the results is often inconsistent owing to heterogeneity in genetic backgrounds. Taking advantage of the fact that organoids can be cultured to retain their physiological functions and that mouse-derived organoids have a defined genetic background, we previously

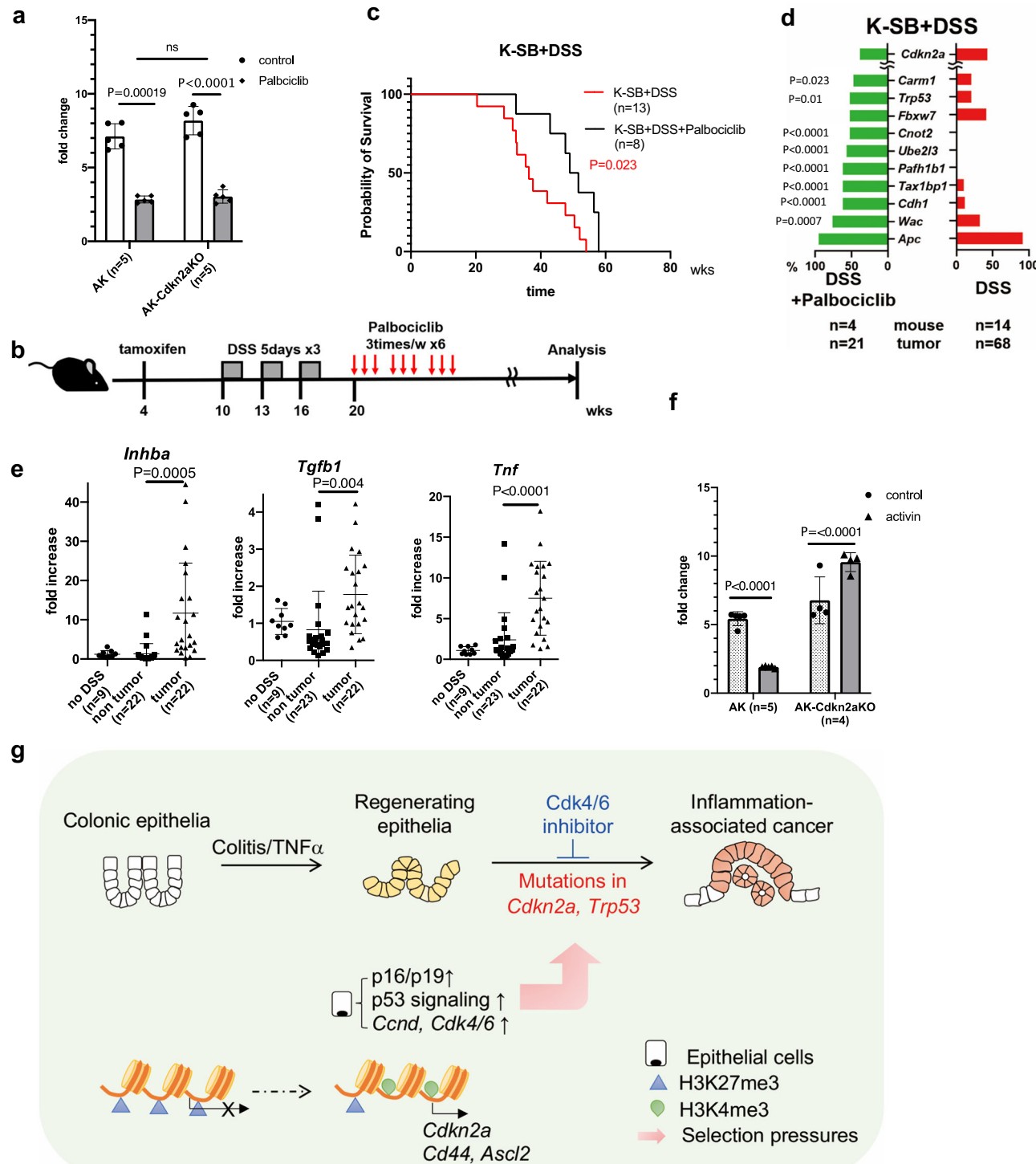

**Fig. 5 | Palbociclib was effective in K-SB + DSS mice. a** Cell viability of AK or AK-Cdkn2aKO organoids in the absence or presence of Palbociclib. Mean values +/- SD, two-way ANOVA. Five independent samples. **b** A picture showing the time course of Palbociclib administration using K-SB + DSS mice. **c** The survival curve of K-SB + DSS mice and K-SB + DSS mice administrated with Palbociclib. Log-rank test. **d** The frequency of insertional mutations for top-10 genes in tumors of K-SB + DSS+Palbociclib mice, and K-SB + DSS mice. Two-sided Fisher's exact test. **e** qPCR analyses for the RNA expression of *Inhba, Tgfb1,* and *Tnf* in normal colon tissues from K-SB mice, non-tumor colon tissues from K-SB + DSS and KT-SB + DSS mice, and tumor tissues from K-SB + DSS and KT-SB + DSS mice. Mean values +/− SD, two-sided *t* test. **f** Cell viability was measured in the presence or absence of activin for AK organoids and AK-Cdkn2aKO organoids, Mean values +/− SD, two-way ANOVA. Five (AK) and four (AK-Cdkn2aKO) independent samples. **g** A graphical outline of this study.

established an experimental system to validate the function of cancer driver genes using mouse tumor-derived AK organoids[44]. AK organoids are not transplantable subcutaneously in immunodeficient mice but become transplantable when driver mutation(s) are introduced[44]. We knocked out *Arhgap5* and *Mecom* in AK organoids by CRISPR-Cas9 using two different gRNAs and confirmed the knockout efficiency (Supplementary Fig. 9a, b). The frequency of subcutaneous tumor development in NSG mice was significantly increased in organoids

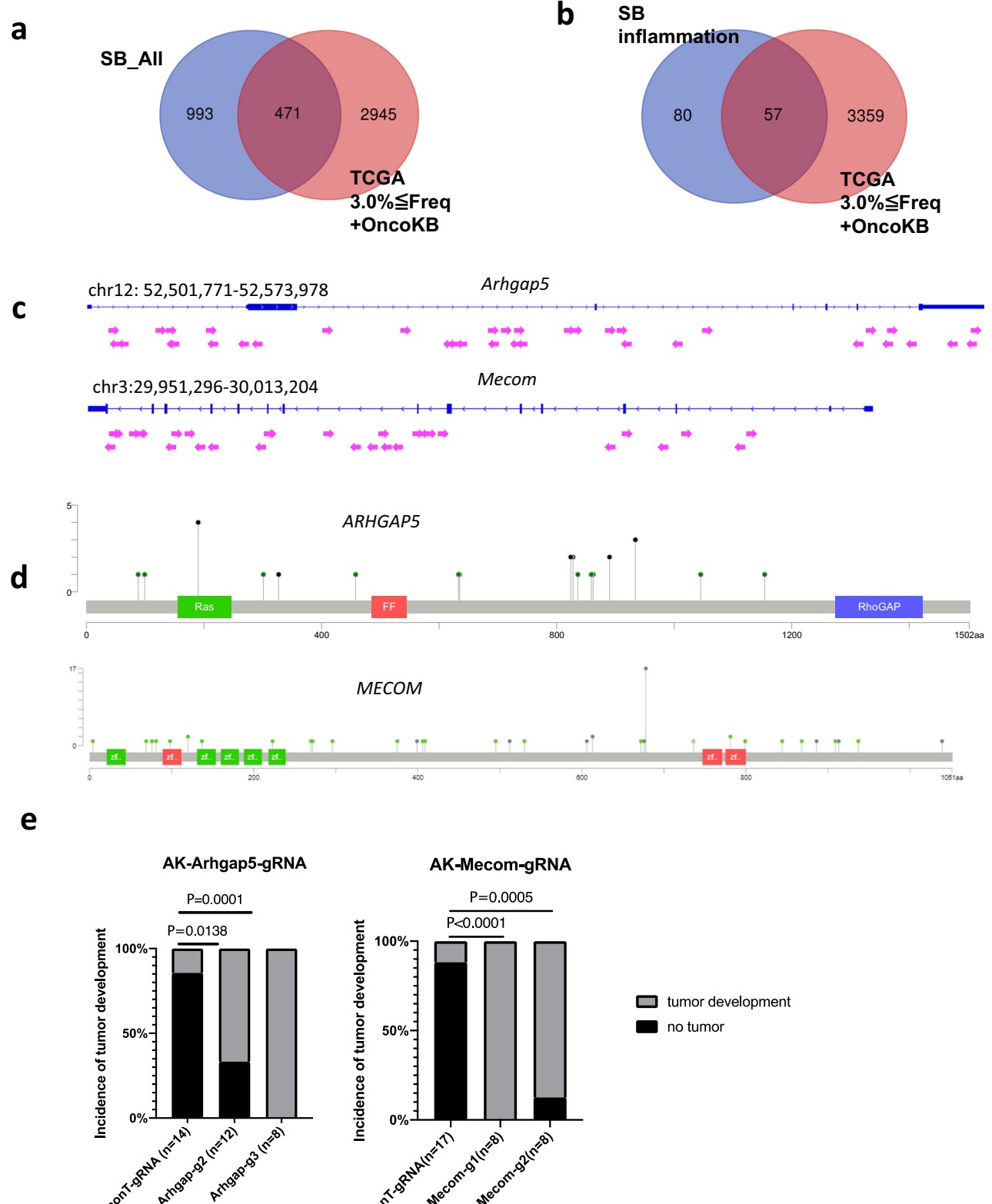

**Fig. 6 | Validation of candidate cancer genes using organoids. a**, **b** A cross-species comparison between the TCGA dataset (3%≦Freq.) combined with OncoKB, and SB screens in this study. **c** The transposon insertion patterns in *Arhgap5* and *Mecom*. Arrows indicate transposon insertions. Arrows pointing to the right mean the direction of the MSCV promoter within the transposon is the same as the direction of transcription. **d** The mutation profiles of *ARHGAP5* and *MECOM* in the TCGA dataset of CRC. **e** Graphs showing the incidence of subcutaneous tumors derived from AK-Cas9-nonT-gRNA (nonT-gRNA), AK-Cas9-Arhgap5-gRNA2 (Arhgap5-g2), and AK-Cas9-Arhgap5-gRNA3 (Arhgap5-g3), AK-Cas9-nonT-gRNA (nonT-gRNA), AK-Cas9-Mecom-gRNA1 (Mecom-g1) and AK-Cas9-Mecom-gRNA2 (Mecom-g2). Data are tested with two-sided Fisher's exact test.

carrying gRNAs targeting *Arhgap5* or *Mecom* compared with that in the non-target gRNA control (Fig. 6e), showing that the two genes are tumor suppressors.

## Discussion

In the present study, we showed that the inflammatory micro-environment exerts unique selective pressures on colon cancer cells and influences the direction of cancer genome evolution (Fig. 5g). One possibility for the frequent insertional mutations in senescence-related genes in inflammation-associated CRC is that TNFα can enhance the plasticity of colonic epithelial cells, leading them in a cell state in which senescence and stemness signals are simultaneously activated. TNFα plays important roles in inflammatory responses and is a pleiotropic cytokine that can induce apoptosis, cell proliferation, and senescence[48]. As TNFα is overexpressed in patients with UC and promotes CAC[49], anti-TNFα therapy has been administered to UC patients.

The present study raised the possibility that *CDKN2A* inactivation could be an early event in inflammation-associated CRC, which is consistent with previous studies showing that the methylation of p14 and p16 is a relatively common early event in UC-associated carcinogenesis[35,36]. In contrast, our study also showed that *CDKN2A* inactivation could be a late event since *Cdkn2a* inactivation was frequently observed in larger tumors, as shown in Fig. 2e. In sporadic CRC, methylation of *CDKN2A* was significantly more frequent in colon cancers with a higher tumor grade and lymph node metastasis[37,38]. The inactivation of *CDKN2A* in the late stage of CRC development may be related to the timing of TNF expression; however, to our knowledge, the association between TNFα expression and *CDKN2A* inactivation in malignant CRC remains unclear and requires further investigation.

Currently, palbociclib is clinically used to treat breast cancer. As we have shown, it could be effective against inflammation-associated CRC with frequent inactivation in senescence-related genes. Several clinical trials using CDK4/6 inhibitors to treat CRC are ongoing. A randomized phase II trial of MEK and CDK4/6 inhibitors in metastatic *KRAS/NRAS* mutant CRC showed no substantial improvement compared to the control groups, although MEK and CDK4/6 inhibitors improved the progression-free survival in the subgroup of CRC[50]. Our study may help to stratify patients for CRC treatment with CDK4/6 inhibitors.

Recent cancer genome sequencing studies have identified a large number of mutated genes. These genes must be distinguished from passenger genes and their function as driver genes must be examined to explore different therapeutic targets. For example, *Arhgap5* encodes the Rho GTPase activating protein, p190BRhoGAP, which negatively regulates Rho GTPase activity in vivo[51]. As Rho GTPase regulates cell cycle progression, Rho GTPase inhibitors may be effective against CRC carrying *ARHGAP5* loss-of-function mutations. Such efforts to functionally validate genes identified from both SB screens and human CRC genome sequencing studies should be continued in the future, as they will facilitate the development of personalized therapies for CRC.

## Methods

All protocols for animal experiments were reviewed and approved by the institutional animal care and use committee of National Cancer Center (Study number: T19-006-M07). The study protocol for experiments on human-derived samples was approved by the Ethics Committees of the National Cancer Center (Study number: 2020-393, 2008-097). Written informed consent was obtained in advance for participants, and no compensation was paid.

### Mice

Rosa26-lsl-SB11[13] and T2/Onc2 (6113) transgenic mice[17] were obtained from Drs. Neal G. Copeland and Nancy A. Jenkins. These strains are also deposited in the NCI mouse repository. Villin-CreERT2 mice were obtained from Dr. Sylvie Robine[18]. Lsl-KrasG12D knock-in mice, Tgfbr2-flox mice, and lsl-Trp53R270H knock-in mice were obtained from the NCI mouse repository. We provided the exact strain, sex, number and age of mice in every experiment in the "Source Data File". Compound mutant mice carrying Rosa26-lsl-SB11, T2/Onc2 (6113), Villin-CreERT2 and sensitizing mutation(s) were treated with 2 mg of tamoxifen (Sigma) dissolved in Corn oil by i.p. for 3 days when they reached 4 weeks of age. To induce colitis, 2.5% Dextran Sodium Sulfate MW 36,000–50,000 (DSS, MP Biochemicals #160110) dissolved in drinking water was administered to mice for 5 days at the age of 10 weeks, then after an interval of 2 weeks, another 2.5% DSS were administered for 5 days. In total, three cycles of DSS administration were performed. When mice showed signs of sickness, such as bleeding, diarrhea, hunched posture, and anemia, we performed necropsies. Mice were opened and the colon was taken out. We counted the number of colonic tumors and measured the diameter. For large tumors whose diameter was ≥3 mm, we obtained both DNA samples and histo-pathological samples. For tumors <3 mm, whole tumors were used for DNA samples. Palbociclib (1.3 mg/mouse/day) was administered three times/weeks for 6 weeks when mice reached 20 weeks of age. The macroscopic appearance of the colon tissues and tumors was taken using a stereoscopic microscope (LEICA M205 FCA) with a LEICA DFC 7000 T camera (Leica). Mouse survivals were calculated by Prism 8 (GraphPad).

The endpoint of the experiment for mice that develop colorectal tumors is when anemia or decreased activity is observed, and this experiment follows the protocol.

### Transplantation

Organoids were dissociated by TrypLE (ThermoFisher) or Accumax (Innovative Cell Technologies). $1 \times 10^5$ cells/100 µl of 50% Matrigel in DMEM/F12 were transplanted subcutaneously to NSG female mice (Charles River) as described previously[44]. Four weeks later, mice were euthanized, and subcutaneous tumors were collected, measured, and processed with HE staining. The animal experimental protocol defines that subcutaneous tumor size should not exceed 10% of the mouse body weight. In our experiment, the tumor size was not exceeded.

### Histology

Tumor tissues were fixed in 4% paraformaldehyde for overnight, and processed for paraffin blocks. Four-micron sections were made and stained by hematoxylin and eosin.

For immunofluorescence staining, 4 µm paraffine sections were deparaffinized, blocked with 3% Donkey serum for 1 hour at r.t., reacted with anti-F4/80 (1/100, Bio-Rad, MCA497RT) overnight at 4 °C, and Donkey anti-Rat IgG (H + L)−488 secondary Ab. Sections were mounted by Antifade Mounting Medium (BECTASHIELD, H-1200).

Histological photos and immunofluorescent photos were taken by IX73 inverted microscope equipped with DP80 digital camera using CellSens software (Olympus). For lower magnification pictures, we used Keyence BZ-X810 unit.

### SB library and NGS

For the preparation of SB libraries, we followed the protocol previously described[16]. Genomes were digested with NlaIII or BfaI and ligated with NlaIII linkers or BfaI linkers. Linker ligated genomes were PCR amplified using IRDR(R1): GCTTGTGGAAGGCTACTCGAAATGTTTGACCC or IRDR(L1): CTGGAATTTTCCAAGCTGTTTAAAGGCACAGTCAAC, and Linker primer: GTAATACGACTCACTATAGGGC. The second PCR was performed using IRDR_nested_illumina:TCGTCGGCAGCGTCAGAT GTGTATAAGAGACAGGTGTATGTAAACTTCCGACTTCAAC, Linker-nested Illumina: GTCTCGTGGGCTCGGAGATGTGTATAAGAGACAGAG GGCTCCGCTTAAGGGAC. PCR products were labeled using a Nextera

XT Index Kit v2 Set A and B (Illumina), and sequenced on a sequencer (Illumina).

## Establishment of an informatics framework and identification of CCGs

All sequencing reads were aligned to the mouse reference genome (NCBI38/mm10) with the SB transposon sequence (T2/Onc2) using BWA (bwa mem -T 0)[52]. Reads with the soft-clipping sequence were extracted since those reads might align with the SB sequence and the genome sequence. Reads with the same boundary site of soft-clipping were gathered as the PCR duplicates and the consensus sequence was made with a simple majority rules algorithm at each base position. If no nucleotide majority reached above the minimum threshold (0.8), the position was considered undefined, and an 'N' was placed at that position in the read. Consensus sequences that consist of ≥3 read depth were extracted. The insertion sites were detected from the boundary sites, adjusting the site using the following criteria: (1) The length of the soft-clipping has to be 22 to 30 bases. (2) Calculating the alignment score of the soft-clipping sequence and 27 bases of the SB sequence next to the 'TA' using the smith waterman algorithm and the score ratio divided the alignment score by soft-clipping length, the score ratio has to be greater than or equal to 0.90. (3) At least 8 bases of soft-clipping sequence next to the TA dinucleotides have to be completely matched to the SB sequence 'TTCAACTG'. Resultant insertion sites from two sequence libraries for each sample were merged. In cases where an insertion site was identified in two libraries, this insertion site was defined as one event.

## Identification of common insertion sites

Common insertion sites (CISs) were analyzed using Poisson distribution statistics. The Poisson distribution calculated the probability of the number of insertion events within a 10 kb window region sliding 5 kb, the expected frequency of insertion events based on the total number of insertions, the size of the genome, and the number of samples. The regions with a $P$ value ≤ 0.001 were collected as CISs and the overlapped regions were stitched as one CIS. The regions with a $P$ value ≤ 0.01 was also applied for Fig. 1d.

position_start: nucleotide position where a CIS starts

position_end: nucleotide position where a CIS ends

count: the number of insertion sites in each 10 kb window

$P$ value: $P$ values in each 10 kb window

minimum $P$ value: the smallest $P$ value within a CIS

SimpleRepeat_score: the score for a simple repeat within a CIS calculated by Ensembl (https://www.ensembl.org/index.html)

simpleRepeat_seq: the simple repeat sequence within a CIS shown in Ensembl insertion_count: the total number of transposon insertions within a CIS

position_count: the total number of unique insertion sites within a CIS

position: the nucleotide position of transposon unique insertion sites in a CIS

gene: gene(s) within a CIS | the type of DNA strand where the gene locates. When two or more genes were detected, each gene was separated by '; '.

sb_gene_direction: Directions of transposons in a gene

human_homolog: corresponding human orthologue(s) for gene(s) within a CIS

Freq_in_tcgaColRec: frequency of gene mutations in the TCGA dataset for CRC

Is_cancer_gene: consensus for cancer driver gene.

## Establishment of mouse colonic organoids

The mouse colon was excised, opened, and washed with PBS. A half portion of the colon was incubated in PBS containing 5 mM EDTA and P/S, for ~1 h on ice on the shaker. Chop the colon with scissors and further mince with a blade. Put in Advanced DMEM/F12 containing HEPES, Glutamax, and pipette vigorously by a P-1000 pipettman to strip crypts. Remove debris with a 100 µm strainer. Centrifuge at $600 \times g$ for 3 min to collect isolated crypts, then embed with 30 µl of Matrigel in the 48-well plate, and overlay 300 µl of medium.

## Identification of inflammation-associated tumor genes

To identify "inflammation-associated tumor genes", the number of tumors with insertional mutations for each CCDG was compared between DSS tumors (K-SB + DSS, KT-SB + DSS, KP-SB + DSS, and P-SB + DSS were all combined) and no DSS tumors (K-SB, KT-SB, KP-SB, and P-SB were combined) by Fisher's exact test ($P < 0.05$). We also identified genes more frequently mutated in DSS tumors than in no DSS tumors in each screen (e.g., K-SB vs K-SB + DSS) using Fisher's exact test ($P < 0.005$) was applied to enrich genotype-specific genes.

## Analyses using the TCGA datasets

For comparison with human datasets with CCDGs, the dataset for TCGA PanCancer Atlas for colorectal adenocarcinoma which included the information on OncoKB was downloaded from cBioPortal. Genes mutated in ≥1.5% of TCGA-CRC were 8905 and compared with 1459 CCDGs. Genes described as "Cancer genes" in the OncoKB database were 1035 and compared with 1459 CCDGs. $P$ values were calculated by Fisher's exact test as described in the previous paper[13].

To see the correlation between TP53 mutations and cytokine expression, the dataset for Pan-cancer analysis of whole genomes (ICGC/TCGA, Nature 2020) was downloaded. The cut-off for the expression of *TNF* and *IFNG* was Z = ±2. The "high" samples were defined when samples showed high mRNA expression (Z ≥2) and/or copy number amplification (AMP). Since there were few samples in which the expression of these cytokines was Z≤−2, or the genomic loci of these cytokines were deleted, we considered the samples other than "high" as "low". For the statistical analysis, Fisher's exact test was used to compare the frequency of *TP53* mutations in "high" and "low" samples. For *TP53* genetic status, mutations (missense mutations, truncation mutations, splice mutations, and deep deletion) were selected and used for comparison.

## Mouse organoid cultures

Wt organoid medium: The medium for wt, KrasG12D/+, KrasG12D/+:p53R270H/+ organoids contained 50% of L-WRN (CRL-3276™, ATCC) conditioned medium, 20% of fetal bovine serum (Biosera), penicillin/streptomycin (ThermoFisher, 1/100), GlutaMax (ThermoFisher, 1/100), Y-27632 (final 10 nM), A83-01 (final 5 nM), SB203580 (final 10 µM), Gastrin (final 10 nM), N-Acetyl-L Cysteine (final 1 µM) and CHIR9902 (final 5 nM) in advanced DMEM-F12.

mTNFα (final 40 ng/ml, Wako), mIL6 (final 100 ng/ml, Wako), mIL-1β (final 10 ng/ml, Wako) and mIFNγ (final 1 ng/ml, Wako) were added to the organoid culture medium.

To activate CreERT2-mediated recombination, we treated organoids with 1 µM 4-hydroxytamoxifen (SIGMA) dissolved in the wt organoid medium for 6 days. The organoid genome was purified by LaboPass Blood Mini (COSMO GENENTECH), and recombination efficiency was determined by a standard PCR protocol.

To knock out genes by CRISPR-Cas9 in wt or KrasG12D/+ organoids, we cloned gRNA into plentiCRISPR_v2 (Addgene, #52961) and generated lentivirus. Sequences for gRNAs were described in Supplementary Table 3. The lentivirus dissolved in Transdux and Enhancer (System Bioscience) was mixed with organoids and placed onto the Matrigel which was polymerized in wells in advance. To confirm whether mutations were introduced by CRISPR-Cas9, we PCR amplified the loci, cloned into pCR™4-TOPO® TA vector (ThermoFisher #450030), and performed Sanger sequencing by M13F or M13R universal primers.

The medium for mouse tumor organoids contained 10 mM HEPES, Penicillin/Streptomycin, N2 supplement (1/100, ThermoFisher), B27 supplement (1/50, ThermoFisher), 50 ng/ml of mEGF (Wako), 5 nM of A83-01, 1 mM of N-Acetyl-L Cysteine in Advanced DMEM-F12 (ThermoFisher), in addition, Y-27632 (final 10 nM) and CHIR9902 (final 5 nM) are added at the time of passage. To pass organoids, TrypLE (ThermoFisher) or Cell recovery solution (Corning) was used, following the manufacturer's protocol. AK organoids were obtained from Dr. Oshima[53]. The establishment of AK-Cas9 organoids was described in our previous report.

To generate knockout in AK organoids, gRNAs[54] were cloned into the BbsI-digested pKLV2-U6gRNA5(Bbsl)-PGKpuro2ABFP-W (Addgene, #67974) lenthi-vector. Lentivirus was generated by transfection of plasmids (pLP1, pLP2, pVSVG, and gRNA-cloned lentivector) into 293FT. Lentiviruses were concentrated by Lenti-X™ Concentrator (TAKARA) and then transduced into AK-Cas9 organoids to knockout genes. Organoids carrying gRNAs were selected by puromycin. Sequences for gRNA were described in the Supplementary Table 3.

## Human organoid cultures

Biopsies of the colonic epithelial mucosa were taken from patients who gave written consent to participate in the study. The sample was collected from 2 Japanese male patients by the surgery. CRC20 organoids were sequenced using the NCC oncopanel v4.

To establish organoids from the normal colonic mucosa, wash biopsy samples with 'Washing medium' (Advanced DMEM containing Glutamax (1/100), 10 mM HEPES, Penicillin/Streptomycin (1/100), Gentamicin (final 50 µl/ml), and 10% FBS), chop with scissors. Put in 10 mM EDTA/PBS and incubate for 1-2 hours on ice with continuous shaking. Remove debris with a 100 µm strainer. Centrifuge at 1,000 g for 3 min to collect isolated crypts, then embed with 30 µl of Matrigel (Corning) or Geltrex (Thermo) in the 48-well plate, and overlay 300 µl of 'Wt organoid medium' containing Gentamicin (final 50 µl/ml) and Plasmocin. 'Wt organoid medium' is described in the mouse organoid section. For maintenance of organoids, we used cell recovery solution to separate organoids from the gel, and pass 1–2 times/week at a ratio of 1:2 or 1:3. TNFα (final 40 ng/ml, Wako) was added to the organoid culture medium.

The medium for human CRC organoids is the same as the 'Wt organoid medium' described above. We don't usually add Gastrin, N-Acetyl-L Cysteine, and CHIR9902 for human CRC organoids. For the maintenance of organoids, we use TrypLE to separate organoids from the gel and pass 1–2 times/week at a ratio of 1:2 or 1:3.

## Organoid growth assay

Remove media from wells in 48-well plates, wash with PBS, add 100 µl of Cell recovery solution (Corning), shaking for 30 min on ice. Add 100 µl of CellTiter-Gro (Promega), vortex for 5 min, incubate for 15 min, r.t. Transfer the mixture to a 96-well plate, and measure luminescence by a micro-plate reader (Agilent BioTek, Synergy H1).

Photographs of the organoids per well were taken at 89 or 96-µm intervals along the z-axis by BZ-X810 (Keyence). These images were merged into a single image in full focus mode by BZ-X800 Analyzer (Keyence). Using 'Hybrid Cell Count' mode of BZ-X800 Analyzer software, the number and area of organoids in the designated area in a well was measured. To remove the image of debris, we removed the area <114 µm$^2$.

To label organoids with EdU, we used Click-iT EdU imaging Kits Alexa Fluor 555 (C 10338, ThermScientific) and followed the manufacturer's protocol. To detect EdU-positive cells, we took images using BZ-X810. Using 'Hybrid Cell Count' mode of BZ-X800 Analyzer software, the number of EdU-positive cells and Hoechst-positive cells were counted.

## qPCR

RNA was purified from organoids using ISOGEN (Nippon Gene) and treated with DNaseI. cDNA was synthesized using 200-600 ng of RNA by the PrimeScript RT Reagent Kit (Takara) following the manufacturer's protocol. Primers were referenced to PrimerBank (https://pga.mgh.harvard.edu/primerbank/) or designed by Primer3 (https://bioinfo.ut.ee/primer3-0.4.0/). Sequences for qPCR primers were described in Supplementary Table 1. Detailed information on primer sequences was described in the supplementary document. qPCR was performed using SYBR Premix Ex Taq, TB Green® Premix Dimer-Eraser™ (Takara) on CFX96 (Bio-Rad) or QuantStudio 3 (ThermoFisher). The fold change was calculated using the ΔΔCt method. At least three independent experiments using three or more samples were performed to see the reproducibility.

## RNA sequencing

Organoid RNA was collected using ISOGEN (Nippon Gene) and treated with DNaseI. 1.5 µg of RNA was used for RNA-seq. Reference sequences for mapping were prepared using rsem-prepare-reference of RSEM version 1.3.1 and STAR version 2.7.7a. Sequence reads were trimmed using trim_galore of Trim Galore version 0.6.7 and Cutadapt version 3.5. The processed reads were mapped to the reference sequences and TPM (Transcripts Per Million) was calculated using rsem-calculate-expression of RSEM version 1.3.1 and STAR version 2.7.10a. Gene and isoform TPM values were extracted from the TPM columns of RSEM result files. GRCm39 genome build was used for the RNA-seq analyses. To draw a heatmap, we used Heatmapper (http://www.heatmapper.ca/). For pathway analysis, an online tool, DAVID[55] (https://david.ncifcrf.gov/) was used. The reference number of datasets is GSE217170. For GSEA analysis, the software, GSEA_4.3.2, was used. A Chip platform (Mouse_Gene_Symbol_Remapping_MSigDB.v2023.1.Mm.chip) was downloaded from the GSEA website. Gene sets were obtained from the previous papers[30,31].

## ChIP-seq analysis

Organoids were collected from nine wells in a 48-well plate, by Cell recovery solution (Corning). Organoids were washed with PBS two times, and dissolved in 10 ml of Advanced DMEM/F12. We added 1 ml of Fixative solution (50 mM HEPES-KOH (pH 7.5), 100 mM of NaCl, 1 mM EDTA, 0.5 mM of EGTA, and 1% formalin (ThermoFisher Scientific, #28906)), and rotated for 10 min, at r.t. To stop the reaction, 2.5 M of Glycine was added and mixed for 5 min, at r.t. by a rotator. Centrifuge the sample for 3 min, at 150× g, and remove the supernatant. Wash the sample by adding 1 ml of PBS containing protease inhibitor cocktail (Roche), centrifuge for 3 min, at 45× g, and remove supernatant. Repeat this step, and store the sample at −80 °C.

Chromatin was sonicated as previously described[56]. In total, 5 µg of the sonicated chromatin was incubated with antibodies against H3K4me3 (Millipore, #05-745 R, clone 15-10-E4, rabbit monoclonal, 3 µl) or H3K27me3 (Cell Signaling Technology, #9733 S, clone C36B11, rabbit monoclonal, 5 µl) at 4 °C overnight. Chromatin was collected using Dynabeads Protein G (ThermoFisher Scientific) and then eluted with TE buffer. After reverse cross-link in the eluate using 200 mM NaCl and proteinase K (ThermoFisher Scientific), DNA was isolated by phenol and chloroform extraction and ethanol precipitation. For ChIP-seq analysis, the sequencing library was prepared with GenNext NGS Library Prep Kit (Toyobo) from 5 ng and sequenced using Illumina Hiseq X (Illumina).

For sequencing QC and adapter trimming of all ChIP-seq data, fastp (version 0.20.0) was used. All reads were aligned to the mouse genome GRCm38/mm10 using bwa-mem (version 2.2) with default parameters. The resultant SAM file was converted into a BAM file,

duplicates were removed using SAMtools (version 1.11), and blacklisted regions in the mouse genome were filtered. Peak calling was performed with the input control using MACS2 (version 2.1.0) with default parameters. Peaks that were unique to the control sample or to the TNFα sample were extracted using a bedtools' subtract tool (bedtools version v2.30.0). Annotation of H3K4me3 and H3K27me3 peaks was performed by comparison with the mouse genome GRCm38/mm10 gtf file. The reference number of datasets is GSE221326 (https://www.ncbi.nlm.nih.gov/geo/query/acc.cgi).

### Single-cell RNA sequencing (scRNA-seq)

Organoids were dissociated into single cells using Accumax (Innovative Cell Technologies), and debris and dead cells were removed by MACS separation column (Milteny Biotech). scRNA-seq libraries of the dissociated organoids were prepared using the Chromium Single Cell 3' Library v3 kit (10x Genomics) according to the manufacturer's protocol. We loaded 6000 cells for each sample. The libraries were sequenced on a NextSeq500 (Illumina) with 28 bp (Read1) and 100 bp (Read2) at a depth of >20,000 reads per cell. Sequencing reads were aligned to the mouse reference genome mm10, and expression matrixes were generated using CellRanger v6.12 (10x Genomics). We used t-stochastic neighbor embedding (t-SNE) and uniform manifold approximation and projection (UMAP) for data visualization. 4,081 cells were analyzed for wt (SC13a1), and 3,459 cells were analyzed for wt+TNFα (SC13b1).

### Bisulfite sequencing analysis

Sodium bisulfite modification was performed using 1 μg of genomic DNA using EZ DNA Methylation™ Kit (ZYMO RESEARCH D5001). For sequencing, sodium bisulfite-modified DNA was amplified with primers for the Cdkn2a region (mp16-bsf-F2: ttgtattggggaggaaggagagattt, mp16-bsf-R: actccatactactccaaataactc). The PCR condition was 94 °C (2 m)→{98 °C (10 s)→58 °C (30 s)→68 °C (30 s)}x45 cycles. KOD DNA Polymerase (TOYOBO) was used. The PCR product was separated by electrophoresis on a 1% agarose gel and extracted using Monarch® DNA Gel Extraction Kit (BioLabs® Inc. #T1020S). dA was attached to both ends of the PCR product using 10x A-attachment mix (TOYOBO TAK-301). The A-attached PCR product was cloned into pCR™4-TOPO® TA vector (ThermoFisher #450030), and sequenced by the M13R universal primer. A ligation reaction was performed for 45 minutes. Approximately ten clones were sequenced using an ABI PRISM 310 sequencer (PE Biosystems, Foster City, CA). To visualize data, QUMA (http://quma.cdb.riken.jp/top/quma_main_j.html) was used.

### Statistics and reproducibility

Experiments were independently performed at least three times for Figs. 3a, g and 4b. No statistical method was used to predetermine sample size, but sample size was set to obtain the maximum results with the minimum number of samples to perform statistical analyses. No data were excluded from the analyses. The investigators were not blinded to allocation during experiments and outcome assessment.

### Reporting summary

Further information on research design and method is available in the Nature Portfolio Reporting Summary linked to this article.

### Data availability

The RNA-seq data and the ChIP-seq data generated in this study have been deposited in the Gene Expression Omnibus (GEO) database under accession codes GSE217170, and GSE221326, respectively. The scRNA-seq data was deposited in the DDBJ under accession codes SAMD00641686 and SAMD00641687. We use OncoKB database [https://www.oncokb.org/cancerGenes], NCBI38/mm10 [https://www.ncbi.nlm.nih.gov/datasets/genome/GCF_000001635.20/] for analyses. Source data are provided with this paper.

### Code availability

The code of SB integration site analysis is deposited in the Github (https://github.com/ni606/sb); https://doi.org/10.5281/zenodo.8330045.

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

## Acknowledgements

We thank Yukari Shiotani for making HE sections, Drs. H. Mano and T. Ushijima for critical comments of the manuscript. We thank Hideki Nishikawa (Keyence Corporation) for assistance with the microscopic analyses. This study was supported by FOREST (JPMJFR2164), JSPS (21K19421, 20H03522), the National Cancer Center Research and Development Fund (2020-A-5), Princess Takamatsu Cancer Fund, Takeda Science Foundation,JH (2022-B-02), and AMED (23ama221529h0001).

## Author contributions

K.S. performed the main experiments, analyzed the data, and contributed to the preparation of the manuscript. N. Hattori performed the ChIP-seq experiment and analyzed the data. N.I. and Y.S. established the informatics pipeline to analyze the SB data. Y.M. performed the DNA methylation experiment and analyzed the data. K.S. established the mouse organoids and assisted in the mouse experiments. D.N. and M.K. analyzed the RNA-seq data. Y.A., N. Hama and T.S. performed scRNA-seq and analyzed the data. K.O. and H. Takamaru collected the human samples and established organoids. H. Takeda designed the experiment, analyzed the data, and wrote the manuscript.

## Competing interests

The authors declare no competing interests. D.N. is currently employed by Chugai Pharmaceutical Co.
