## [Peer Review File · Nature Communications]

Sleeping Beauty transposon mutagenesis identified genes and pathways involved in inflammation-associated colon tumor developmentEditorial Note: Parts of this Peer Review File have been redacted as indicated to maintain the confidentiality of unpublished data.

REVIEWER COMMENTS

Reviewer #1 (Remarks to the Author):

The manuscript 'Colon tumor cells are prone to inactivate the cellular senescence pathway in the inflammatory cancer microenvironment' recently submitted to Nat Commun made a series of efforts to provide mechanistic findings of CRC development in the setting of inactivated senescence. The authors identified 142 genes by in vivo screening, including senescence and TGF β -activin signaling genes, as more often altered in inflammation-associated KrasG12D colon tumors. They found that in response to the inflammatory cytokine, colon epithelial cells simultaneously were able to engage both stemness and senescence signaling by enhancing cell plasticity. This could act as a selection pressure to mutate senescence genes for inflammation-associated CRC, for which the Cdk4/6 inhibitor was effective in vivo.

The authors concluded how important the inactivation of senescence pathways for CRC development and progression in the inflammatory microenvironment is, and they believe these findings can lead to the potential achievement of precision medicine in future cancer research. Generally, this study has presented some new datasets and raised a number of important questions for cancers especially CRC. However, the vast majority of these data are too preliminary and lack sufficient mechanistic depth. Experimental design was incomplete for quite a few assays, and data analysis largely failed to get the major points. Together, the current manuscript is not ready to be considered for publication in the journal, with much more efforts required for future improvement.

Minor points:

1. The title is not well generated, and clearly suffers from the lack of specific findings such as genes and/or mechanisms involved in CRC.
2. Abstract. 'Furthermore, we functionally validated Arhgap5 and Mecom, with unknown significance mutations in CRC and inflammation-associated CRC, were tumor-suppressor genes, providing novel possible therapeutic targets for inflammation-associated CRC.' This part is confusing, with blurry description of what the authors have found. How about inflammation-unassociated CRC, do these genes (Arhgap5 and Mecom) also have implications? What do unknown significance mutations mean?
3. Page 3. 'Conversely, different orders are implicated in inflammation-associated tumors, indicating that inflammation acts as a unique selection pressure for malignant progression.' Different orders of what? From the context prior to this sentence, it is hard to make the conclusion.
4. Page 3. 'However, most of the recurrently mutated genes in inflammation-associated cancers differed between reports, due to large individual differences in genetic backgrounds and the degree and duration of inflammation in patient samples, therefore, the molecular mechanisms of how inflammation impacts the acquisition of gene mutations remain largely unknown.' There lacks appropriate citations for this part, or these are completely derived from the authors' speculation.
5. Page 3, the last part of the 1st paragraph, '...how we treat inflammation-associated CRC, using model systems with controlled inflammatory conditions and defined genetic backgrounds.' This part needs to be rewritten to avoid misunderstanding.
6. The last part of Page 4. 'We thus provided new datasets that will help developing new

personalized therapies for treating inflammation-associated CRC, and a new concept that cellular plasticity in the inflammatory microenvironment could act as evolutionary pressures driving CRC development and progression.' Grammar errors.

7. Page 5. The authors stated 'The number of large colonic tumors (≥ 5 mm) were frequently observed in K-SB-DSS and KT-SB-DSS, while no large tumors were developed in their controls, suggesting that colitis accelerated tumor progression, which could cause shortened mouse survivals.' The conclusion was rather bold and needs to be more prudent, even with further data acquired.

8. Fig 2D. Lower p-values indicated stronger selection for the CISs, showing that strong cooperation with KrasG12D and Cdkn2a mutations in inflammation-associated tumor development. Did the authors make comparison with the controls?

9. Fig 2E. ...the Cdkn2a mutation frequency was significantly higher in DSS tumors, clearly indicating the positive selection for Cdkn2a mutations under the inflammatory microenvironment in the early stage of tumor development. The conclusion seems to be problematic and should be carefully drawn.

10. Fig. 3D and Supplementary Fig. 3D, E. In DSS organoids, activation of senescence signaling and TNF α signaling was also observed by RNA-seq. Did the authors want to address that activation of senescence signaling was driven by functionally engaged TNF α ?

11. Page 9. Beyond the fact that transcriptome analysis indicated the upregulation of senescence signaling genes such as Cdkn2a, Mapk11, Mapk12 and Slc25A4, there should be further demonstration that TNF α induces senescence signaling in the colonic epithelial cells, such as elimination of this inflammatory cytokine in the experimental settings, rather than directly making such a conclusion.

12. There are numerous grammatical problems throughout the manuscript, which were not paid attention by authors at all.

Reviewer #2 (Remarks to the Author):

This manuscript describes experiments designed to determine what genes and genetic pathways control colorectal cancer (CRC) development in the context of gastrointestinal tract (GI) inflammation. It is known that chronic GI tract inflammation, such as occurs in colitis syndromes, can predispose to CRC in human patients. Many pathways have been proposed to account for this including dysbiosis, loss of mucin, altered stromal to epithelial cell signaling and other factors. It is also known that the somatically acquired CRC mutations in the background of ulcerative colitis differ from those seen in other CRC cases. By using a mouse model of inflammation induced CRC susceptibility and a Sleeping Beauty (SB) mediated transposon mutagenesis system, the authors provide data from a well-controlled model system to provide additional insight into this process. Specifically, the authors used SB mutagenesis in a dextran sodium sulfate (DSS)-induced CRC model. DSS has been classically combined with mutagenic chemicals to create CRC in mice on a background of inflammation. The authors convincingly show that DSS treatment accelerates CRC development on several predisposed backgrounds. They decided to focus on genetic analyses of CRC-like tumors that developed in the colon to focus their attention on the most human-relevant cases. These were all good choices. The applicants discovered genes more likely to be altered by transposon insertion specifically in DSS treated cases, including Trp53 and Cdkn2a. These data are like human in that TP53 altered cases are found more often in colitis associated cancer (CAC) than in other CRCs. The enrichment for Cdkn2a mutation was evident even in small tumors and became more pronounced with CRC progression (i.e. larger tumors). The authors used human and mouse organoid model systems to study the

effects of altering CDKN2A on sensitivity to the effects of inflammation-induced TNF α expression, showing this allows cells to proliferate in response to TNF α , which itself induces expression of CDKN2A. In addition, some excellent ChIPseq and scRNAseq data were generated detailing the effects of TNF α on colon organoids, which support their contention that it induces senescence and stemness. The authors also tested a CDK4/6 inhibitor, Palbociclib for effects on Cdkn2a KO organoids in vitro and on mice in vivo. The results suggest use of such an inhibitor for inflammation induced CRC. The authors also showed that Cdkn2a KO organoids were resistant to the growth suppressive effects of activin, which induced by inflammation. Finally, the applicants also functionally validated 2 additional TSG candidates from their screen, Mecom and Arhgap5, by knocking them out in Kras/Cdkn2a mutant organoids and transplanting them subcutaneously. These data do not really drive the overall hypothesis or conclusions of the paper. But, these data do emphasize the fact that SB mutagenesis identified bona fide cancer genes. Overall, the authors presented evidence for their main conclusion that TNF- α in the chronic inflammatory microenvironment provides strong selective pressure for CRC-initiating cells to obtain Cdkn2a inactivation, and that CDK4/6 inhibitors could be used as a therapeutic option in CRC developing after inflammation, as a result. These data could help patient stratification for CRC treatment with agents, such as CDK4/6 inhibitors in cases of inflammation-associated disease. While the manuscript is overall well written and the data are of generally very high quality and completeness, certain key major evidence as well as minor adjustments are required. The manuscript does really report a lot of data overall. Specific comments follow:

Major Comments:

- λ Is the percentage of invasive adenocarcinomas really increased in DSS groups compared to controls? This issue seems to be glossed over as panel J only shows DSS cases.
- λ What assays were performed to show DSS had the intended effect of inducing inflammation? Were controls compared?
- λ There is a lack of direct evidence on TNF- α being the selection pressure that leads to CDKN2A inactivation. As an important piece of evidence to support the main hypothesis, Fig. 4P and 4Q missed time-matched negative control. While the findings on Fig. 4H and 4J could somehow support the conclusion, but the results here could not reconcile with the previous findings in Fig. 2C, in which KrasG12D-driven SB mice did obtain more CDKN2A mutations in DSS-treated group. Compelling evidence will be creating a transgenic mouse model with CDKN2A mutation and treat with or without DSS, or create CDKN2A-KO human CRC organoid clones and treat with or without TNF- α . Or at least show that TNF- α -treated KrasG12D-Cdkn2a KO and/or KrasG12D-Trp53R270H organoids showed higher cell cycle activity.
- λ Popivanova (J Clin Invest. 2008) had demonstrated that TNF- α is associated with CRC initiation and development under chronic colitis microenvironment. The author should include this study into introduction.
- λ Does TNF α treatment of human organoids result in methylation and decreases in gene expression?
- λ It is unclear how the authors prioritized genes to discover a list of 142 genes to be inflammation associated tumor genes (lines 316-320).

Minor comments:

- λ In places the text is awkward or needs revision. A proof reading is required. For example, lines 165-167 makes little sense. The comment on lines 300-302 is vague (resistant in what way?). Lines 324 and 325 mentions two genes, but then lines 325-327 mentions 3 genes.

What is the 3rd gene?

λ The authors should cite work describing CDK4/6 inhibitors in CRC treatment – trials are underway already.

λ It'd helpful to perhaps better explain the nomenclature used in lines 86 and 87 for mouse lines.

λ Figure 1C is difficult to read for color blind readers. It might be necessary to split the data into separate figures also.

λ There is insufficient evidence and reference supporting the conclusion of CDKN2A inactivation as an early event in human CRC. Based on Fig. 2E, in which SB-induced CDKN2A mutation was found to generate more tumors, the authors reasoned that CDKN2A inactivation could be a preferred for CRC initiation under chronic inflammatory microenvironment. However, this conclusion should be further supported by more experiments or human database analysis – perhaps on polyps? Is such data available? Based on current data, TNF- α -induced CDKN2A inactivation in tumor and organoids could also be a late event. The authors should also discuss about how their findings fit/contradict to previous studies on the timing of CDKN2A inactivation in CRC.

λ Provide the rationale of the cutoffs set for TNG and IFNG, and also provide the correlation between CDKN2A mutation status and TNG and IFNG expression in Fig. 4N and 4O.

λ Unify presentation style for organoid proliferation data for Fig. 4J, 4K, 5A, and 5F.

λ Two-way ANOVA should be used for analyzing data from Fig. 5A and 5F.

Reviewer #3 (Remarks to the Author):

The authors use an in vivo SB-mediated mutagenesis screening tool to investigate mutations linked to inflammation-associated colorectal carcinogenesis, giving rise to interesting data and a high-quality study. Combining in vivo work, bioinformatics and organoid work, the authors are able to elegantly address the selective pressures that might correspond to what intestinal (stem) cells encounter in inflammatory bowel diseases. The description and interpretation of the results need to be strengthened, as detailed below.

Major:

- Cdkn2a is referred to as a reliable marker for senescence. Although its role in cytosclerosis is undisputed, it is also simply a cell cycle gene, and a TGFB target gene. Authors should show in their organoid experiments (Fig 3) that persistent senescence is indeed triggered using bonafide markers. If this is the case, authors should explain how functional senescence would allow for the derivation and propagation of organoids.

- In case persistent cytosclerosis or senescence can be functionally validated more convincingly, authors should explain why a key part of this phenotype cannot be simply explained by inflammation-mediated initiation of the TGFB pathway?

- Relatedly, the cell state described herein should be compared and contrasted to LRC/Mex3a/+4 crypt-like stem cells [see e.g. PMID: 35773527], as well as against YAP-mediated fetal progenitor/revival/regenerating cell states — to be better able to appreciate distinction and novelty (as claimed, line 355).

Moderate/minor:

- Please revise the abstract. For instance, in line 16, it is unclear which cytokine is meant; lines 21–23 are hard to follow or contain grammatical errors.

- In the main text, there are some minor issues with grammar or English style, such as line 272 and 371: please revise.
- Method description, data visualization, and statistics can use some improvement. Specifically, I noticed that for the p value in line 130, it is not clear what exactly is tested here (or how). For some of the data plotted (e.g. in Fig 1D, 3C, H, K, and M), there is no mention of the n, or individual data points are not plotted.
- Also, the methods/statistics description for TCGA/OncoKB database analysis seems to be missing. In this case (lines 306–320), how can an overlap be statistically significant?
- I would rephrase lines 135–141; looking at two genes in this set of genetic backgrounds makes ‘showing unique genetic selection processes depending on pre-existing mutations’ sound somewhat overstated.
- Conclusion lines 98–100, should include " in the presence of transposon activity and mutated driver genes". Relatedly, I think the authors should, at some point, explain the choice not to include SB-negative control mouse lines.
- Lines 142–152 amount to a circular argument and flawed logic. I don't see a convincing reason to rule out the possibility that inflammation elicits a tissue response (including stroma) that drives higher proliferation as an early oncogenic driver.
- Lines 165–167 make no sense to me.
- In Fig 3P, no difference is shown for the methylation patterns for Cdkn2a. Clarify and revise please.
- Line 246–247: While the two characteristics may have a statistically significant negative correlation, clearly they are not precisely mutually exclusive. Also, I could not find the methods on this TCGA data use, e.g. how is low/high defined, what statistical test was used.
- The difference between the experimental setup/data in Fig 4Q/R and S8A are not clear to me.

Reviewer #1 (Remarks to the Author):

The manuscript ‘Colon tumor cells are prone to inactivate the cellular senescence pathway in the inflammatory cancer microenvironment’ recently submitted to Nat Commun made a series of efforts to provide mechanistic findings of CRC development in the setting of inactivated senescence. The authors identified 142 genes by in vivo screening, including senescence and TGFb-activin signaling genes, as more often altered in inflammation-associated KrasG12D colon tumors. They found that in response to the inflammatory cytokine, colon epithelial cells simultaneously were able to engage both stemness and senescence signaling by enhancing cell plasticity. This could act as a selection pressure to mutate senescence genes for inflammation-associated CRC, for which the Cdk4/6 inhibitor was effective in vivo.

The authors concluded how important the inactivation of senescence pathways for CRC development and progression in the inflammatory microenvironment is, and they believe these findings can lead to the potential achievement of precision medicine in future cancer research. Generally, this study has presented some new datasets and raised a number of important questions for cancers especially CRC. However, the vast majority of these data are too preliminary and lack sufficient mechanistic depth. Experimental design was incomplete for quite a few assays, and data analysis largely failed to get the major points. Together, the current manuscript is not ready to be considered for publication in the journal, with much more efforts required for future improvement.

Minor points:

1. The title is not well generated, and clearly suffers from the lack of specific findings such as genes and/or mechanisms involved in CRC.

As per the reviewer’s suggestion, we have changed the title to “Sleeping Beauty transposon mutagenesis identified genes and pathways involved in inflammation-associated colon tumor development” in the revised manuscript.

2. Abstract. ‘Furthermore, we functionally validated Arhgap5 and Mecom, with unknown significance mutations in CRC and inflammation-associated CRC, were tumor-suppressor genes, providing novel possible therapeutic targets for inflammation-associated CRC.’ This part is confusing, with blurry description of what the authors have found. How about inflammation-unassociated CRC, do these genes (Arhgap5 and Mecom) also have implications? What do unknown significance mutations mean?

We apologize for the insufficient explanation for *Arhgap5* and *Mecom*. In our SB screens, these two genes mutated in both no DSS and DSS tumors but more frequently mutated in DSS tumors. There are no reports describing the mutation in *Arhgap5* and *Mecom* in human inflammation-associated CRC, however, mutations (the majority are stop mutations) are observed in the TCGA dataset for human sporadic CRC. Since the significance of the mutation or the function of these two genes in human sporadic CRC development is unknown, we functionally validated and showed that the two genes are tumor suppressor genes.

We have corrected the entire abstract.

3. Page 3. ‘Conversely, different orders are implicated in inflammation-associated tumors, indicating that inflammation acts as a unique selection pressure for malignant progression.’ Different orders of what? From the context prior to this sentence, it is hard to make the conclusion.

We have revised the sentence and added the following sentence in lines 25-28 of the revised manuscript.

“Chronic inflammation is known to promote CRC development and progression, and a subtype of CRC characterized by high inflammatory signatures correlates with poor prognosis^{2,3}. Patients with inflammatory bowel disease (IBD) are at a higher risk of developing CRC^{4,5,6}.”

4. Page 3. ‘However, most of the recurrently mutated genes in inflammation-associated cancers differed between reports, due to large individual differences in genetic backgrounds and the degree and duration of inflammation in patient samples, therefore, the molecular mechanisms of how inflammation impacts the acquisition of gene mutations remain largely unknown.’ There lacks appropriate citations for this part, or these are completely derived from the authors’ speculation.

We have revised the sentence and explained results of genomic analyses in colitis-associated cancers in previous studies with appropriate references in lines 28-32 of the revised manuscript. “Genomic analysis of colitis-associated cancers (CAC) has shown that *TP53* is the most commonly mutated gene in CAC⁷⁻¹⁰. In addition, several genes, such as *SOX9*⁷, *EP300*⁷, *ARID1A*⁹, *CDH2*⁹, and *FBXW7*¹⁰ were more frequently mutated in CAC. These data suggested that the genetic mutational profile of CAC differs from that of sporadic CRC⁷⁻¹⁰.”

5. Page 3, the last part of the 1st paragraph, ‘...how we treat inflammation-associated CRC, using model systems with controlled inflammatory conditions and defined genetic backgrounds.’ This part needs to be rewritten to avoid misunderstanding.

We have changed the sentence to ‘To comprehensively understand the genes and pathways involved in inflammation-associated CRC, we performed *Sleeping Beauty* (SB) transposon screening in a mouse model of colitis’ in lines 33-35 of the revised manuscript.

6. The last part of Page 4. ‘We thus provided new datasets that will help developing new personalized therapies for treating inflammation-associated CRC, and a new concept that cellular plasticity in the inflammatory microenvironment could act as evolutionary pressures driving CRC development and progression.’ Grammar errors.

We have corrected the sentence to “Thus, we have provided new datasets that will help develop new personalized therapies for treating inflammation-associated CRC and a novel concept that cellular plasticity in the inflammatory microenvironment could act as an evolutionary pressure driving CRC development” in lines 52-55 of the revised manuscript.

7. Page 5. The authors stated ‘The number of large colonic tumors (≥ 5 mm) were frequently observed in K-SB-DSS and KT-SB-DSS, while no large tumors were developed in their controls, suggesting that colitis accelerated tumor progression, which could cause shortened mouse survivals.’ The conclusion was rather bold and needs to be more prudent, even with further data acquired.

We have changed the sentence to “Large colonic tumors (≥ 5 mm) were frequently observed in K-SB+DSS and KT-SB+DSS, mice, whereas no large tumors were developed in their controls, suggesting that colitis promoted large tumor development.” In addition, we changed the description of DSS-treated mice to +DSS (e.g., K-SB mice treated with DSS were shown as K-SB+DSS), in lines 75-77 of the revised manuscript.

8. Fig 2D. Lower p-values indicated stronger selection for the CISs, showing that strong

cooperation with *Kras*G12D and *Cdkn2a* mutations in inflammation-associated tumor development. Did the authors make comparison with the controls?

The p-values for *Cdkn2a* and *Trp53* in no DSS control (left) and DSS tumors (right, also in Fig. 2D) are shown below. These data showed that the p-values for two genes were lower in DSS tumors than in those of control groups for all genotypes, indicating the enrichment of insertional mutations for *Cdkn2a* and *Trp53* in DSS tumors. We added the data in Supplementary Fig. 2B.

[Redacted]

To calculate the p-value for each CIS, we counted unique transposon insertions within a 10-kb window and compared the insertion frequency in that region with the frequency that would be expected if all insertions in the chromosome in each screen were randomly distributed.

9. Fig 2E. ...the *Cdkn2a* mutation frequency was significantly higher in DSS tumors, clearly indicating the positive selection for *Cdkn2a* mutations under the inflammatory microenvironment in the early stage of tumor development. The conclusion seems to be problematic and should be carefully drawn.

We have changed the sentence to “... the frequency of *Cdkn2a* mutations was significantly higher in DSS tumors (Fig. 2E). Furthermore, the proportion of tumors with *Cdkn2a* mutations increased in adenocarcinomas (Fig. 2F). These data suggest that colonic cells carrying a *Kras*G12D mutation selectively acquire *Cdkn2a* mutations to promote tumor development in the inflammatory microenvironment” in lines 125-129 of the revised manuscript.

10. Fig. 3D and Supplementary Fig. 3D, E. In DSS organoids, activation of senescence signaling and TNF α signaling was also observed by RNA-seq. Did the authors want to address that activation of senescence signaling was driven by functionally engaged TNF α ?

Fig. 3E (previous Fig. 3D) and Supplementary Fig. 3D, E show that the expression profile in DSS organoids is different from that in wt organoids. We think the change in the cell state of colonic epithelial cells is already induced by DSS *in vivo* since we see the upregulation of *Tnf* and *Cdkn2a* in the DSS-induced colitis tissue (Supplementary Figure 3C), therefore, DSS organoids may partially retain the cell state that was acquired *in vivo*. DSS changes the cellular state of colonic epithelial cells in mice as previously reported (Yui et al., Cell Stem Cell, 2018).

On the other hand, as the reviewer suggests, TNF α produced from DSS organoids may have activated cellular senescence signaling in the culture dish. We checked the RNA-seq data to compare the count of *Tnf* mRNA and found that the TPM was 17 and 27 for wt organoids and DSS organoids, respectively. The basal level of *Tnf* expression was observed in wt organoids, and the expression was 1.5 times more in DSS organoids. These data suggest that it may also be possible to induce senescence signaling activation by TNF α produced from DSS organoids.

To describe more clearly what we would like to say with the analyses, we have changed the sentence to “we performed RNA-seq using DSS and control organoids and found different expression profiles as characterized by the activation of senescence signaling in DSS organoids (Fig. 3D, Supplementary Fig. 3D, E)” in lines 139-141 of the revised manuscript.

11. Page 9. Beyond the fact that transcriptome analysis indicated the upregulation of senescence signaling genes such as *Cdkn2a*, *Mapk11*, *Mapk12* and *Slc25A4*, there should be further demonstration that TNF α induces senescence signaling in the colonic epithelial cells, such as elimination of this inflammatory cytokine in the experimental settings, rather than directly making such a conclusion.

In previous studies, it has been shown that blocking TNF α signaling in the colitis mouse model considerably reduces inflammation, as observed by decreased infiltration of macrophages and neutrophils (Popivanova *et al.*, JCI, 2008), and decreased expression of inflammatory cytokines/chemokines, such as *Il6* and *Ccl2* (Blais *et al.*, Int. J. Mol. Sci., 2023). These data indicate that TNF α regulates several cytokine signals in colitis and that it is difficult to compare colitis with and without TNF α , not affecting other cytokine expressions. Therefore, we decided to use organoids to see the effects of TNF α *in vitro*.

12. There are numerous grammatical problems throughout the manuscript, which were not paid attention by authors at all.

We have carefully revised the manuscript and sent it to an English editing service.

Reviewer #2 (Remarks to the Author):

This manuscript describes experiments designed to determine what genes and genetic pathways control colorectal cancer (CRC) development in the context of gastrointestinal tract (GI) inflammation. It is known that chronic GI tract inflammation, such as occurs in colitis syndromes, can predispose to CRC in human patients. Many pathways have been proposed to account for this including dysbiosis, loss of mucin, altered stromal to epithelial cell signaling and other factors. It is also known that the somatically acquired CRC mutations in the background of ulcerative colitis differ from those seen in other CRC cases. By using a mouse model of inflammation induced CRC susceptibility and a Sleeping Beauty (SB) mediated transposon mutagenesis system, the authors provide data from a well-controlled model system to provide additional insight into this process. Specifically, the authors used SB mutagenesis in a dextran sodium sulfate (DSS)-induced CRC model. DSS has been classically combined with mutagenic chemicals to create CRC in mice on a background of inflammation. The authors convincingly show that DSS treatment accelerates CRC development on several predisposed backgrounds. They decided to focus on genetic analyses of CRC-like tumors that developed in the colon to focus their attention on the most human-relevant cases. These were all good choices. The applicants discovered genes more likely to be altered by transposon insertion specifically in DSS treated cases, including *Trp53* and *Cdkn2a*. These data are like human in that TP53 altered cases are found more often in colitis associated cancer (CAC) than in other CRCs. The enrichment for *Cdkn2a* mutation was evident even in small tumors and became more pronounced with CRC progression (i.e. larger tumors). The authors used human and mouse organoid model systems to study the effects of altering CDKN2A on sensitivity to the effects of inflammation-induced TNF α expression, showing this allows cells to proliferate in response to TNF α , which itself induces expression of CDKN2A. In addition, some excellent ChIPseq and scRNAseq data were generated detailing the effects of TNF α on colon organoids, which support their contention that it induces senescence and stemness. The authors also tested a CDK4/6 inhibitor, Palbociclib for effects on *Cdkn2a* KO organoids in vitro

and on mice in vivo. The results suggest use of such an inhibitor for inflammation induced CRC. The authors also showed that Cdkn2a KO organoids were resistant to the growth suppressive effects of activin, which induced by inflammation. Finally, the applicants also functionally validated 2 additional TSG candidates from their screen, Mecom and Arhgap5, by knocking them out in Kras/Cdkn2a mutant organoids and transplanting them subcutaneously. These data do not really drive the overall hypothesis or conclusions of the paper. But, these data do emphasize the fact that SB mutagenesis identified bona fide cancer genes. Overall, the authors presented evidence for their main conclusion that TNF- α in the chronic inflammatory microenvironment provides strong selective pressure for CRC-initiating cells to obtain Cdkn2a inactivation, and that CDK4/6 inhibitors could be used as a therapeutic option in CRC developing after inflammation, as a result. These data could help patient stratification for CRC treatment with agents, such as CDK4/6 inhibitors in cases of inflammation-associated disease. While the manuscript is overall well written and the data are of generally very high quality and completeness, certain key major evidence as well as minor adjustments are required. The manuscript does really report a lot of data overall. Specific comments follow:

We thank the reviewer for the suggestions that we feel have helped us to improve our manuscript and for the statement that “these data could help patient stratification for CRC treatment with agents, such as CDK4/6 inhibitors in cases of inflammation-associated disease”. We are really encouraged by the reviewer’s statement.

Major Comments:

1. Is the percentage of invasive adenocarcinomas really increased in DSS groups compared to controls? This issue seems to be glossed over as panel J only shows DSS cases.

Thank you for the comment. Since we had few tumors large (>3mm) enough to make HE sections for no-DSS control mice, we could not perform a statistical comparison. We have toned down the sentence to “colitis promoted the development of large tumors” in lines 76-77 of the revised manuscript.

2. What assays were performed to show DSS had the intended effect of inducing inflammation? Were controls compared?

To set up the experimental condition to induce inflammation by DSS, we referred to a couple of papers (Grivennikov et al., Cancer Cell, 2009; Oshima et al., Can Res 2015). To confirm

whether DSS induced inflammation, we chronologically made HE sections after DSS for the colon of DSS-treated mice and age-matched control mice. We confirmed that mucosal damage, ulcer, and immune cell infiltration were observed only in DSS-treated mice and that inflammation lasted almost seven days. In addition, we stained the sections with an antibody for a macrophage marker, F4/80, and showed that macrophages accumulated in the damaged colonic mucosa in DSS-treated mice as shown in Supplementary Figure 3A. Furthermore, we performed real-time PCR and confirmed that inflammatory cytokines, such as TNF α and Il6, were induced in the DSS-treated colon as shown in Supplementary Figure 3C. Based on these data, we concluded that DSS treatment had the intended effect of inducing inflammation in the colon.

3. There is a lack of direct evidence on TNF- α being the selection pressure that leads to CDKN2A inactivation. As an important piece of evidence to support the main hypothesis, Fig. 4P and 4Q missed time-matched negative control. While the findings on Fig. 4H and 4J could somehow support the conclusion, but the results here could not reconcile with the previous findings in Fig. 2C, in which KrasG12D-driven SB mice did obtain more CDKN2A mutations in DSS-treated group. Compelling evidence will be creating a transgenic mouse model with CDKN2A mutation and treat with or without DSS, or create CDKN2A-KO human CRC organoid clones and treat with or without TNF- α . Or at least show that TNF- α -treated KrasG12D-Cdkn2a KO and/or KrasG12D-Trp53R270H organoids showed higher cell cycle activity.

For new Fig. 4S and T (previous Fig. 4Q and R, I think the reviewer indicates Fig. 4Q and R, not 4P and 4Q), the data on the left were time-matched controls. The organoids were divided into two at the passage, and cultured with or without TNF α for 3-4 months side by side, and genomic DNA was collected and analyzed. We apologize for an inadequate description that might have led to some misunderstanding. We have revised the picture in Fig. 4R and changed the label for Fig. 4S and Fig. 4T.

To show that TNF α -treated KrasG12D-Cdkn2aKO and/or KrasG12D-Trp53R270H showed higher cell cycle activity, we performed additional experiments using 2 different approaches.

1. We quantified the area of organoids on day 1 and day 3 from the passage. We found organoids with TNF α expanded faster than organoids without TNF α , showing TNF α promoted proliferation of organoids carrying mutations in *Cdkn2a* or *Trp53*. This approach is used to quantify organoid proliferation as shown in a recent paper (Alvarez-Varela *et al.*, Nature Cancer, 2022). We included the data in Figures 4L and M.

Material and Methods: Photographs of the organoids per well were taken at 89 or 96- μm intervals along the z-axis by BZ-X810 (Keyence). These images were merged into a single image in full focus mode by BZ-X800 Analyzer (Keyence). Using 'Hybrid Cell Count' mode of BZ-X800 Analyzer software, the number and area of organoids in the designated area in a well were measured. To remove the image of debris, we removed the area $<114 \mu\text{m}^2$.

2. Alternatively, we quantified the DNA synthesis by EdU labeling for KrasG12D-Cdkn2aKO with or without TNF α . The proportion of EdU-positive cells in Hoechst-positive cells was significantly higher in organoids with TNF α , showing that TNF α promoted the cell cycle activity in KrasG12D-Cdkn2aKO organoids. We included the result in Supp Fig. 7C.

Material and Methods: To label organoids with EdU, we used Click-iT EdU imaging Kits Alexa Fluor 555 (C 10338, ThermoScientific) and followed the manufacturer's protocol. To detect EdU-positive cells, we took images using BZ-X810 (Keyence). Using the 'Hybrid Cell Count' mode of BZ-X800 Analyzer software, the number of EdU-positive cells and Hoechst positive cells were counted.

[Redacted]

4. Popivanova (J Clin Invest. 2008) had demonstrated that TNF- α is associated with CRC initiation and development under chronic colitis microenvironment. The author should include this study into introduction.

Thank you for your helpful suggestion. We included the paper in the discussion to explain the role of TNF α in the colitis-associated CRC.

5. Does TNF α treatment of human organoids result in methylation and decreases in gene expression?

We have not tried the long-term culture of normal human organoids with or without TNF α , so we do not have answers to the question. We believe that the same result as what we see in the mouse organoids will be obtained from human organoids because p14 expression was increased in the presence of TNF α .

6. It is unclear how the authors prioritized genes to discover a list of 142 genes to be inflammation associated tumor genes (lines 316-320).

We described how to identify “inflammation-associated tumor genes” in the materials and methods section (lines 608-614) as follows:

“To identify “inflammation-associated tumor genes”, the number of tumors with insertional mutations for each CCDG was compared between DSS tumors (K-SB+DSS, KT-SB+DSS, KP-SB +DSS, and P-SB+DSS were all combined), and no DSS tumors (K-SB, KT-SB, KP-SB, and P-SB were combined) by Fisher's exact test ($P < 0.05$). We also identified genes more frequently mutated in DSS tumors than in no DSS tumors in each screen (e.g., K-SB vs K-SB+DSS) using Fisher's exact test ($P < 0.005$) was applied to enrich genotype-specific genes.”

We also included a brief description in in lines 99-101 of the revised manuscript:

“To identify genes that were more frequently mutated in DSS tumors, we compared the insertional mutation frequencies between DSS and non-DSS tumors (Supplementary Fig. 2) and enriched 142 genes as “inflammation-associated genes” (Supplementary Table 3).”

Minor comments:

7. In places the text is awkward or needs revision. A proof reading is required. For example, lines 165-167 makes little sense. The comment on lines 300-302 is vague (resistant in what way?). Lines 324 and 325 mentions two genes, but then lines 325-327 mentions 3 genes. What is the 3rd gene?

We have carefully revised the manuscript and sent it to an English editing service.

Lines 165-167 were changed to “Notably, several *Hoxb* genes were upregulated in the DSS organoids (Fig. 3E, Supplementary Fig. 3F). The increased expression of *Hoxb* correlated with the demethylation of histone H3 lysine-27 trimethylation (H3K27me3) in the promoter²⁶, implicating a change in histone modifications in DSS organoids” in lines 141-145 of the revised manuscript.

Lines 300-302 were corrected to “Interestingly, AK-Cdkn2aKO organoids were resistant to activin-induced growth arrest (Fig. 5F). These data suggest that inactivation of *Cdkn2a* may confer cancer cells with the ability to survive in an activin-rich inflammatory environment (Fig. 5G)” in lines 281-284 of the revised manuscript.

Lines 325-327 were corrected to “The transposon insertions appeared to inactivate these two genes (Fig. 6C)” in lines 307-308 of the revised manuscript.

8. The authors should cite work describing CDK4/6 inhibitors in CRC treatment – trials are underway already.

Thank you for your valuable information. We have added the sentence below in the discussion and included the reference.

“Several clinical trials using CDK4/6 inhibitors to treat CRC are ongoing. A randomized phase II trial of MEK and CDK4/6 inhibitors in metastatic *KRAS/NRAS* mutant CRC showed no substantial improvement compared to the control groups, although MEK and CDK4/6 inhibitors improved the progression-free survival in the subgroup of CRC⁵¹. Our study may help to stratify patients for CRC treatment with CDK4/6 inhibitors” in lines 354-359 of the revised manuscript.

9. It'd helpful to perhaps better explain the nomenclature used in lines 86 and 87 for mouse lines.

We added the sentence “DSS-treated mice are shown as +DSS (e.g., K-SB mice treated with DSS are shown as K-SB+DSS).” In line 69 in the revised manuscript.

10. Figure 1C is difficult to read for color blind readers. It might be necessary to split the data into separate figures also.

We changed the color for survival curves and separated the data by genotype. These survival data were shown separately in Figure 1C and Supplementary Figure 1A.

11. There is insufficient evidence and reference supporting the conclusion of CDKN2A inactivation as an early event in human CRC. Based on Fig. 2E, in which SB-induced CDKN2A mutation was found to generate more tumors, the authors reasoned that CDKN2A inactivation could be a preferred for CRC initiation under chronic inflammatory microenvironment. However, this conclusion should be further supported by more experiments or human database analysis – perhaps on polyps? Is such data available? Based on current data, TNF- α -induced CDKN2A inactivation in tumor and organoids could also be a late event. The authors should also discuss about how their findings fit/contradict to previous studies on the timing of CDKN2A inactivation in CRC.

Fig. 2E shows that the percentage of nascent tumors carrying *Cdkn2a* insertional mutations is significantly higher in DSS tumors compared to no-DSS tumors, implicating that *Cdkn2a* inactivation occurs relatively early in tumor development under the chronic inflammatory microenvironment in our SB model. We then set up experiments using non-transformed colonic epithelial organoids and showed that organoids carrying mutations in *Kras* and *Cdkn2a* proliferated faster with TNF α than without TNF α . We think our *in vitro* data partially models the initiation step of inflammation-associated tumor development and supports our *in vivo* data. In the previous analysis of patients with ulcerative colitis, p14(ARF) methylation was observed in 19 of 38 (50%) adenocarcinomas, 4 of 12 (33%) dysplasia, and 3 of the 5 (60%) nonneoplastic UC mucosae. In contrast, 3 of 40 (3.7%) normal tissues showed p14(ARF) methylation (chi(2) test: P=0.0003) (Sato *et al.*, Cancer Res., 2002). The study suggests that methylation of p14(ARF) is a relatively common early event in UC-associated carcinogenesis, supporting our finding.

As the reviewer pointed out, CDKN2A inactivation could be also a late event, since *Cdkn2a* inactivation was also frequently observed in larger tumors as shown in Fig. 2E. In sporadic CRC, methylation of CDKN2A was significantly more frequent in colon cancers with a higher tumor grade and lymph node metastasis (Bihl *et al.* Journal of Translational Medicine 2012, 10:173; Psofaki *et al.*, World Journal of Gastroenterology, 2010), which is consistent with our finding. However, there has been no report describing the association between TNF α expression and CDKN2A inactivation in malignant CRC, and future investigation will be required.

We included the sentence below in the discussion in lines 341-351 of the revised manuscript. “The present study raised the possibility that *CDKN2A* inactivation could be an early event in inflammation-associated CRC, which is consistent with previous studies showing that the methylation of p14 and p16 is a relatively common early event in UC-associated carcinogenesis^{35,36}. In contrast, our study also showed that *CDKN2A* inactivation could be a late event since *Cdkn2a* inactivation was frequently observed in larger tumors, as shown in Fig. 2E. In sporadic CRC, methylation of *CDKN2A* was significantly more frequent in colon cancers with a higher tumor grade and lymph node metastasis^{37,38}. The inactivation of *CDKN2A* in the late stage of CRC development may be related to the timing of TNF expression; however, to our knowledge, the association between TNF α expression and *CDKN2A* inactivation in malignant CRC remains unclear and requires further investigation.”

12. Provide the rationale of the cutoffs set for TNG and IFNG, and also provide the correlation between *CDKN2A* mutation status and TNG and IFNG expression in Fig. 4N and 4O.

The cut-off for the expression of *TNF* and *IFNG* was $Z=\pm 2$. We did not see a correlation between *CDKN2A* mutation status and the expression of *TNF* or *IFNG* in this dataset. The result suggested that increased expression of *TNF* could cooperate with *TP53* mutations in many types of cancers, but not with *CDKN2A*. It may be possible that overexpression of *TNF* specifically cooperated with *CDKN2A* loss in *KRAS*-mutated CRC, but we could not confirm this due to the limited sample number in the cBioportal.

13. Unify presentation style for organoid proliferation data for Fig. 4J, 4K, 5A, and 5F.

We have corrected the style for organoids viability assay for Fig. 5A, and 5F.

14. Two-way ANOVA should be used for analyzing data from Fig. 5A and 5F.

As per the reviewer’s suggestion, we analyzed Fig. 5A and 5F using Two-way ANOVA.

Reviewer #3 (Remarks to the Author):

The authors use an in vivo SB-mediated mutagenesis screening tool to investigate mutations linked to inflammation-associated colorectal carcinogenesis, giving rise to interesting data and a high-quality study. Combining in vivo work, bioinformatics and organoid work, the authors are able to elegantly address the selective pressures that might correspond to what intestinal (stem) cells encounter in inflammatory bowel diseases. The description and interpretation of the results need to be strengthened, as detailed below.

Thank you very much for the reviewer's suggestions and comments especially regarding the interpretation of stemness in the organoid work. We believe our manuscript is much improved since the reviewer kindly introduced the important paper describing *Mex3a* that expressed in the cancer stem cells acquiring drug resistance.

Major:

- *Cdkn2a* is referred to as a reliable marker for senescence. Although its role in cytotaxis is undisputed, it is also simply a cell cycle gene, and a TGF β target gene. Authors should show in their organoid experiments (Fig 3) that persistent senescence is indeed triggered using bona fide markers. If this is the case, authors should explain how functional senescence would allow for the derivation and propagation of organoids.

Thank you very much for your helpful comments. We examined whether functional senescence was induced in TNF α -treated organoids by staining organoids with X-Gal to see the β -galactosidase activity, but we did not see prominent signs of senescence induction by the experiment.

To further examine whether the activation of senescence signaling correlated with cytotaxis, we focused on the Ki67(-) cell population using scRNA-seq data. Although the ratio of Ki67(-) cells was almost unaffected by TNF α , Ki67(-):Ascl2 (+) cells that could be non-dividing stem cells were increased five times by TNF α . Furthermore, the activation of senescence signaling as indicated by expression of *Trp53* or *Cdkn2a* or *Cdkn1a* was observed in most of Ki67(-):Ascl2(+) cells. These data have suggested that senescence marker expression and cytotaxis were correlated in stem cell marker positive cells (e.g., Ascl2(+)), and that the cell state of non-dividing cells was greatly affected by TNF α . Since these non-dividing stem cells should have the potential to proliferate in certain conditions, the disruption of the senescence pathway may put these cells in the actively dividing cell state in the presence of TNF α . In contrast, the ratio of Ki67(+) dividing cells is almost unaffected by TNF α , and this is one of the reasons that organoids can proliferate in the presence TNF α . We reflected our analyses in new Fig. 3T and Supplementary Figure 5.

[Redacted]

X-gal staining for KrasG12D organoids and KrasG12D cultured in the presence of $TNF\alpha$ for a month. Wt organoids treated with 5FU for 2 days were used as positive controls. Red arrows were cells stained by X-Gal, indicating that these were senescent cells. Bars: 50 μm

- In case persistent cytostasis or senescence can be functionally validated more convincingly, authors should explain why a key part of this phenotype cannot be simply explained by inflammation-mediated initiation of the TGF β pathway?

As the reviewer has pointed out, it is possible that the inflammatory microenvironment makes cancer cells acquire *Cdkn2a* mutations to evade growth arrest caused by TGF β , since *Cdkn2a* is a known TGF β target gene. However, we couldn't obtain convincing data to support this possibility in our organoid culture condition. In the presence of TGF β , organoids stop proliferating and eventually die even if organoids carry a loss of function mutation in *Apc*, an activating point mutation in *Kras*, and a loss of function mutation in *Cdkn2a*. Also, we did not see significant induction of *Cdkn2a* expression in AK organoids when treated with TGF β . This is one of the reasons why we think acquiring mutations in *Cdkn2a* cannot be explained by the role of TGF β for colonic epithelial cells in the early stage of tumor development.

[Redacted]

- Relatedly, the cell state described herein should be compared and contrasted to LRC/Mex3a/+4 crypt-like stem cells [see e.g. PMID: 35773527], as well as against YAP-mediated fetal progenitor/revival/regenerating cell states — to be better able to appreciate distinction and novelty (as claimed, line 355).

We appreciate the reviewer for this comment. We compared our RNA-seq dataset obtained from TNF α -treated wt organoids with the gene sets used in the two publications (PMID: 35773527 and PMID: 29249464) as suggested by the reviewer. Interestingly, the cell state in the TNF α -treated wt organoids was similar to *Mex3a* positive cells, *Lgr5* positive cells, and fetal progenitor cells, but not correlated with LRC cells or other differentiated cells. These data supported our finding that TNF α could induce stemness in the colonic epithelial cells. We included the data in Fig. 3M and added the sentence below in lines 162-168 of our revised manuscript.

[Redacted]

To further determine which cell types in the intestine are more similar to TNF α -treated organoids, we compared our RNA-seq dataset with previously reported gene sets^{30,31}. Interestingly, the cell state in the TNF α -treated wt organoids was similar to *Mex3a* positive cells, *Lgr5* positive cells, and colitis cells, but not correlated with label-retaining cells (LRC) or other

differentiated cells. These data also supported our finding that TNF α could induce stemness in the colonic epithelial cells (Fig. 3M).

Moderate/minor:

- Please revise the abstract. For instance, in line 16, it is unclear which cytokine is meant; lines 21–23 are hard to follow or contain grammatical errors.

We have corrected the abstract.

- In the main text, there are some minor issues with grammar or English style, such as line 272 and 371: please revise.

We have carefully revised the manuscript and sent it to an English editing service.

- Method description, data visualization, and statistics can use some improvement. Specifically, I noticed that for the p value in line 130, it is not clear what exactly is tested here (or how). For some of the data plotted (e.g. in Fig 1D, 3C, H, K, and M), there is no mention of the n, or individual data points are not plotted.

The p-value in line 130 of the previous manuscript means that the tumor carrying insertional mutations in gene involved in the pathway (i.e., cellular senescence pathway, chromatin remodeling) is more frequent in DSS tumors. We compared the number of tumors with insertional mutations in genes in each pathway between DSS and no DSS tumors by Fisher's exact test. We revised the sentence to "the tumor carrying insertional mutations in senescence or chromatin remodeling genes were significantly more frequent in DSS tumor than in no DSS tumor ($P < 0.01$, by Fisher's exact test)." In line 116-118 of the revised manuscript.

We added the information on the number of samples for Fig 3C, H, K, M, 4E, and individual data points were plotted for all the data including Fig 1D, 3C, H, K, M, 4E, and supplementary figures.

- Also, the methods/statistics description for TCGA/OncoKB database analysis seems to be missing. In this case (lines 306–320), how can an overlap be statistically significant?

We calculated whether the overlap between the genes in the TCGA-CRC dataset and 1,459 genes identified from our SB screening was significant or not by Fisher's exact test, as described in the previous paper (Starr *et al.*, *Science* 2009). This is based on the idea of whether the probability of the overlap between the TCGA and the SB dataset is greater than the probability for an overlap of 2 gene sets randomly extracted from 18,000 human genes. We added the method for the comparison with TCGA datasets in the material and method section as follows.

Analyses using the TCGA datasets

For comparison with human datasets with CCDGs, the dataset for TCGA PanCancer Atlas for colorectal adenocarcinoma which included the information on OncoKB was downloaded from cBioPortal. Genes mutated in $\geq 1.5\%$ of TCGA-CRC were 8,905 and compared with 1,459 CCDGs. Genes described as “Cancer genes” in the OncoKB database were 1,035 and compared with 1,459 CCDGs. P-values were calculated by Fisher’s exact test as described in the previous paper¹³.

- I would rephrase lines 135–141; looking at two genes in this set of genetic backgrounds makes ‘showing unique genetic selection processes depending on pre-existing mutations’ sound somewhat overstated.

We apologize for the description that may have confused the reviewer. We deleted lines 135–141 since what these sentences meant was already described in the sentence preceding it.

- Conclusion lines 98–100, should include " in the presence of transposon activity and mutated driver genes". Relatedly, I think the authors should, at some point, explain the choice not to include SB-negative control mouse lines.

Thank you for the reviewer’s suggestion. As per the reviewer, we added the words “in the presence of SB insertional mutations” in line 87 of the revised manuscript.

We added the following sentence in the revised text. “We did not include DSS-treated SB negative controls since several reports showed that the number of tumors in DSS-treated mice without carcinogens was relatively low^{24,25}.” In lines 69-72 of the revised manuscript.

- Lines 142–152 amount to a circular argument and flawed logic. I don't see a convincing reason to rule out the possibility that inflammation elicits a tissue response (including stroma) that drives higher proliferation as an early oncogenic driver.

We changed the sentence to: “To focus on how *Cdkn2a* mutations are involved in the early stages of tumor development, insertional mutation frequencies in nascent tumors (<2 mm) were compared. Interestingly, in nascent tumors, the frequency of *Cdkn2a* mutations was significantly higher in DSS tumors (Fig. 2E). Furthermore, the proportion of tumors with *Cdkn2a* mutations increased in adenocarcinomas (Fig. 2F). These data suggest that colonic cells

carrying a *Kras*G12D mutation selectively acquire *Cdkn2a* mutations to promote tumor development in the inflammatory microenvironment.” in lines 124-130 of the revised manuscript.

- Lines 165–167 make no sense to me.

We have revised the sentence as follows:

“Notably, several *Hoxb* genes were upregulated in the DSS organoids (Fig. 3E, Supplementary Fig. 3F). The increased expression of *Hoxb* correlated with the demethylation of histone H3 lysine-27 trimethylation (H3K27me3) in the promoter²⁶, implicating a change in histone modifications in DSS organoids.”, which are in lines 140-146 of the revised manuscript.

- In Fig 3P, no difference is shown for the methylation patterns for *Cdkn2a*. Clarify and revise please.

We apologize for the insufficient explanation of the figure. To show the genomic loci where the level of histone methylation was different between wt organoids and wt+TNF α organoids, we added the subtracted peak to show more methylated regions in wt for H3K27me3, and more methylated regions in wt+TNF α for H3K4me3, in new Fig 3Q (previous Fig. 3P). As you can see, the promoter region of p19 was demethylated for H3K27me3 in wt+TNF α organoids, which means the activation of transcription. We also revised supplementary Fig. 4 to include the subtracted peak.

- Line 246–247: While the two characteristics may have a statistically significant negative correlation, clearly they are not precisely mutually exclusive. Also, I could not find the methods on this TCGA data use, e.g. how is low/high defined, what statistical test was used.

The cut-off for the expression of *TNF* and *IFNG* was $Z=\pm 2$. The “high” samples were defined as high mRNA expression ($Z \geq 2$) and/or copy number amplification (AMP). Since there were few samples in which the expression of these cytokines was $Z \leq -2$, or the genomic loci of these cytokines were deleted, we considered the samples other than “high” as “low”. For the statistical analysis, Fisher’s exact test was used to compare the frequency of *TP53* mutations in “high” and “low” samples. We described how we performed the analysis using the TCGA dataset in the material and method section. And we have toned down the sentence in lines 226-

230 of the revised main text as follows “Cancer tissues with *TNF* overexpression/amplification co-occurred with mutations in *TP53* ($P=0.0442$), raising the possibility that increased levels of $TNF\alpha$ are associated with *TP53* mutations in human cancers (Fig. 4P). Alternatively, $IFN\gamma$ overexpression/amplification did not cooperate with *TP53* mutations (a tendency for mutual exclusion, $P=0.0368$) (Fig. 4Q).”

- The difference between the experimental setup/data in Fig 4Q/R and S8A are not clear to me.

We apologize for the insufficient explanation. The data in Fig. 4S, T (previous 4Q/R) and S8A were obtained in the same experiment design, but these were independent experiments/samples. We deleted the Supplementary Fig 8A.

REVIEWERS' COMMENTS

Reviewer #1 (Remarks to the Author):

The authors responded to the vast majority of my concerns, and the manuscript can be considered to proceed.

Reviewer #2 (Remarks to the Author):

The revised manuscript describes experiments designed to lend insight into the molecular pathways that govern inflammation induced colorectal cancer. It uses as a starting point for these studies, a transposon mutagenesis accelerated model of CRC in which loss of function insertion mutations in *Cdkn2a* were selected for more often in a background of chronic inflammation than in control CRC without inflammation as a predisposing background. The authors provide convincing data that TNF α produced in the inflammatory environment exerts an early negative selective pressure characterized by *Cdkn2a* induction and thus selects for early loss of *Cdkn2a* in nascent CRC tumors. The revised manuscript is much improved in clarity and presentation overall with new key references also. Also, new data is provided. These new data show that *Cdkn2a* loss potentiates cell cycling in the presence of TNF α . The authors also better defined the senescent state induced by TNF α . These and other measures improved the manuscript very much.

Reviewer #3 (Remarks to the Author):

The authors have sufficiently addressed my concerns. However, they should check clarity and English language on the changed passages, e.g. the one describing Fig 2C (line 116–7).

2nd Peer Review

Reviewer #3 (Remarks to the Author):

The authors have sufficiently addressed my concerns. However, they should check clarity and English language on the changed passages, e.g. the one describing Fig 2C (line 116–7).

We read the entire manuscript again and made corrections.

Lines 116-117 “the tumors carrying insertional mutations in senescence and chromatin remodeling genes were significantly more frequent in DSS tumor than in no DSS tumor”
→ “the number of tumors carrying insertional mutations in senescence and chromatin remodeling genes was significantly more frequent in DSS tumor than in no DSS tumor”